# Cell-autonomous and non-cell-autonomous effects of Arginase 2 on cardiac aging

**Duilio M Potenza[1]\*, Xin Cheng[1], Guillaume Ajalbert[1], Andrea Brenna[1], Marie-Noelle Giraud[2], Aurelien Frobert[2], Stephane Cook[2], Kirsten D Mertz[3], Zhihong Yang[1]\*, Xiu-Fen Ming[1]\***

[1]Laboratory of Cardiovascular and Aging Research, University of Fribourg, Fribourg, Switzerland; [2]Cardiology, Department of Endocrinology, Metabolism, and Cardiovascular System, Faculty of Science and Medicine, University of Fribourg, Fribourg, Switzerland; [3]Institute for Pathology, Cantonal Hospital, Baar, Switzerland

## eLife Assessment

This study provides **fundamental** information on how Arg-II participates in cardiac aging. The phenotypic data provide **convincing** evidence of non-cell-autonomous contributions to aging-related pathologies. Overall, the study highlights the importance of intercellular signaling in maintaining cardiac health during aging.

\*For correspondence:
duilio.potenza@unifr.ch (DMP);
zhihong.yang@unifr.ch (ZY);
xiu-fen.ming@unifr.ch (X-FM)

**Competing interest:** The authors declare that no competing interests exist.

**Abstract** Aging is a predominant risk factor for heart disease. Aging heart reveals low-grade chronic inflammation, cell apoptosis, cardiac fibrosis, and increased vulnerability to ischemic injury. The underlying molecular mechanisms responsible for cardiac aging and its susceptibility to injury are not fully understood. Although literature reports a role for mitochondrial Arginase 2 (ARG2) in heart failure, contradictory results are reported. How ARG2 participates in cardiac aging is still unknown. In this study, we demonstrate that *Arg2* is not expressed in cardiomyocytes from aged mice and humans but is upregulated in non-myocyte cells, including macrophages, fibroblasts, and endothelial cells. Mice with genetic deficiency of *Arg2* (*Arg2-/-*) are protected from age-associated cardiac inflammation, myocyte apoptosis, interstitial and perivascular fibrosis, endothelial-mesenchymal transition (EndMT), and susceptibility to ischemic injury. Further experiments show that ARG2 mediates IL-1β release from macrophages of old mice, contributing to the cardiac aging phenotype. In addition, ARG2 enhances mitochondrial reactive oxygen species (mtROS) and activates cardiac fibroblasts that is inhibited by inhibition of mtROS. Thus, our study demonstrates a non-cell-autonomous effect of ARG2 on cardiomyocytes, fibroblasts, and endothelial cells mediated by IL-1β from aging macrophages as well as a cell-autonomous effect of ARG2 through mtROS in fibroblasts contributing to cardiac aging phenotype.

## Introduction

Aging is a predominant risk factor for cardiovascular disease (CVD; *Moturi et al., 2022*). Cardiac aging is accompanied by low-grade chronic inflammation termed 'inflammaging' (*Cevenini et al., 2013*), which is linked to gradual development of cardiac fibrosis and heart failure, whereby macrophages, cardiac fibroblasts, and endothelial cells play critical roles in age-associated cardiac remodeling and dysfunction (*Abdellatif et al., 2023*; *Meschiari et al., 2017*). Studies in experimental animal models and in humans provide evidence demonstrating that aging heart is more vulnerable

to stressors such as ischemia/reperfusion injury and myocardial infarction and exhibits less efficient reparative capability after injury as compared to the heart of young individuals (*Bujak et al., 2008*; *Mariani et al., 2000*). Even in the heart of apparently healthy individuals of old age, chronic inflammation, cardiomyocyte senescence, cell apoptosis, interstitial/perivascular tissue fibrosis, endothelial dysfunctio,n and endothelial-mesenchymal transition (EndMT), and cardiac dysfunction either with preserved or reduced ejection fraction rate are observed (*Abdellatif et al., 2023*; *Ruiz-Meana et al., 2020*). Despite the intensive investigation in the past, the underlying causative cellular and molecular mechanisms responsible for the cardiac aging phenotype and the susceptibility of aging heart to injurious stressors are not fully elucidated yet. Moreover, most preclinical studies preferentially use relatively juvenile animal models that hardly recapitulate clinical scenarios. It is therefore critical to investigate mechanisms of cardiac aging phenotype with a naturally advanced aging animal model.

Research in the past years suggests a wide range of functions of the enzyme arginase, including the cytosolic isoenzyme Arginase 1 (ARG1) and the mitochondrial isoenzyme Arginase 2 (ARG2) in organismal aging as well as age-related organ structural remodeling and functional changes (*Caldwell et al., 2015*; *Li et al., 2022*). These functions of arginase have been implicated in cardiac and vascular aging (*Yepuri et al., 2012*) and cardiac ischemia/reperfusion injury in rodents (*Jung et al., 2010*). ARG1, originally found in hepatocytes, plays a crucial role in the urea cycle in the liver to remove ammonia from the blood (*Sin et al., 2015*), whereas ARG2 is inducible in many cell types under pathological conditions, contributing to chronic tissue inflammation and remodeling (*Moretto et al., 2019*). Both enzymes are able to catalyze the divalent cation-dependent hydrolysis of L-arginine (*Ming and Yang, 2013*). Studies in the literature, including our own, demonstrate that ARG2 is involved in the aging process and promotes organ inflammation and fibrosis, and inhibition of arginase or genetic ablation of *Arg2* slows down the aging process and protects against age-associated organ degeneration such as lung (*Zhu et al., 2023*), pancreas (*Xiong et al., 2017b*), and heart (*Xiong et al., 2017a*). However, controversial results on expression of arginase and its isoforms at the cellular levels in the heart are reported. While some studies linked ARG1 to the cardiac ischemia/reperfusion injury (*Jung et al., 2010*), others suggest a role of ARG2 in this context (*Heusch et al., 2010*). Even protective functions of ARG2 are suggested in some studies, based on correlation of ARG2 levels with cardiac injury or systemic administration of ARG2 in rat models (*Huang et al., 2018*; *Lu et al., 2012*). Importantly, cellular localization of arginase and its isoenzyme in the heart, that is expression of arginase in cardiomyocytes and non-cardiomyocytes hase not been systematically analyzed and confirmed. Finally, whether and how ARG2 participates in cardiac aging and whether it is through cell-autonomous and/or paracrine effects are not known.

Therefore, our current study is aimed at investigating which cell types in the heart express arginase and what enzymatic isoforms. How arginase and this isoenzyme in these cells influence the cardiac aging process and enhance susceptibility of the aging heart to ischemia/reperfusion injury.

## Results

### Increased *Arg2* expression in aging heart

In heart from young (3–4 months in age) and old (22–24 months in age) mice, an age-associated increase in *Arg2* expression in both wt male and female mice is observed (*Figure 1A*). This age-associated increase in *Arg2* expression is more pronounced in females than in males (*Figure 1A*). In contrast to *Arg2*, *Arg1* mRNA expression is very low in the heart (with very high Ct values, data not shown), and no age-dependent changes of *Arg1* gene expression are observed in wt and in *Arg2*$^{-/-}$ mice of either gender (*Figure 1—figure supplement 1A and B*), suggesting a specific enhancement of ARG2 in aging heart. Further experiments are thus focused on ARG2 and female mice. We made efforts to analyze ARG2 protein levels by immunoblotting in the heart tissues. However, immunoblotting is not sensitive enough to detect ARG2 protein in the heart tissue of either young or old animals. Therefore, further experiments are performed to detect ARG2 protein at the cellular level with confocal microscopic immunofluorescence staining as described in later sections.

### *Arg2* ablation reduces cardiac aging phenotype

Next, we investigated the impact of *Arg2* ablation on cardiac phenotype. Masson's staining reveals an enhanced peri-vascular and interstitial fibrosis in wt old mice, which is mitigated in age-matched

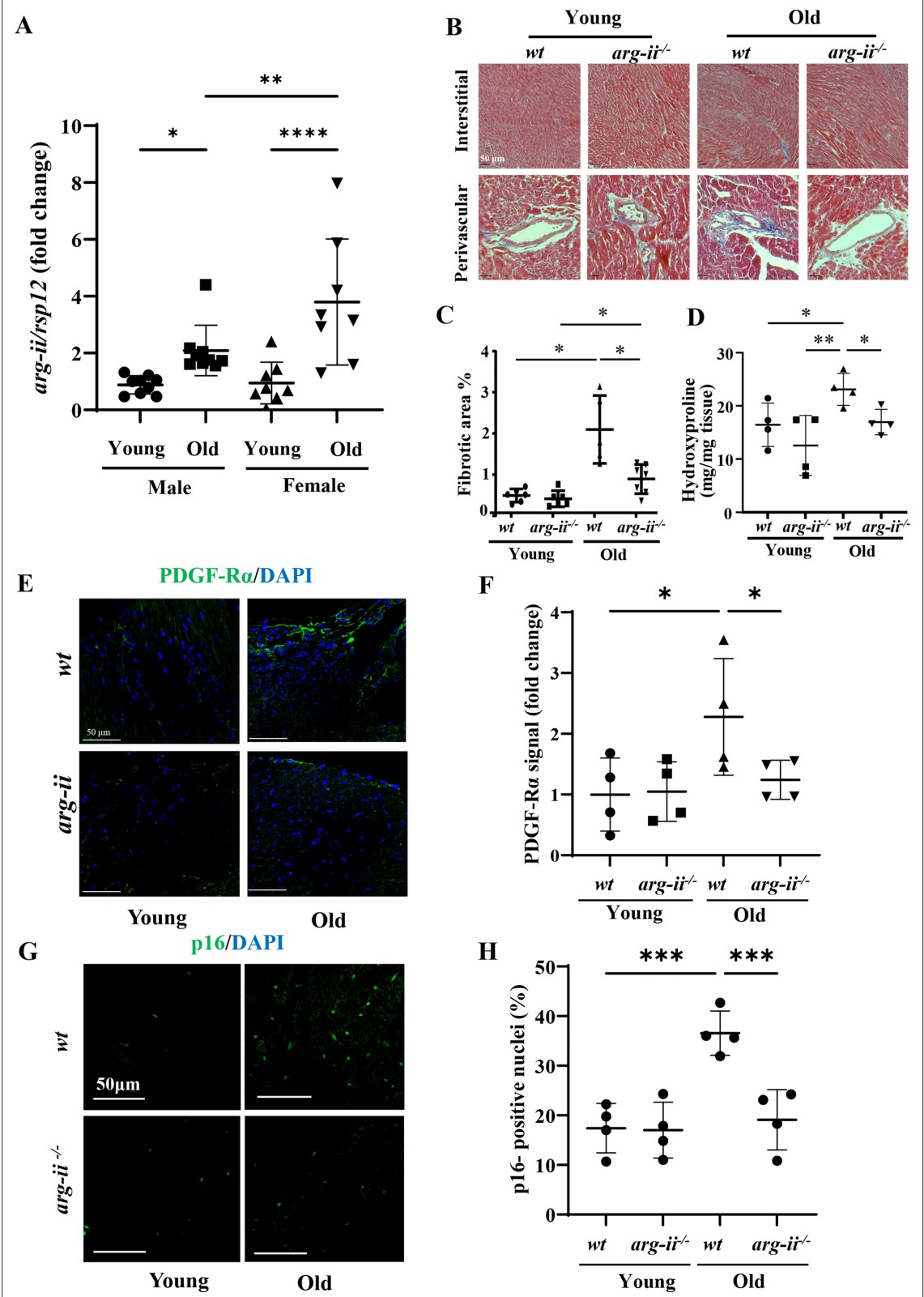

**Figure 1.** Age-associated increase in Arg2 levels in mouse heart. (**A**) *Arg2* mRNA levels of male and female young (3–4 months) and old (20–22 months) wild type (wt) heart tissues analyzed by qRT-PCR. *Rps12* served as the reference (n=8–9 animals per group); (**B**) Representative histological images of heart interstitial and perivascular fibrosis in young and old wt and *Arg2⁻/⁻* female mice. Fibrosis is shown by the blue-colored Trichrome Masson's staining. Scale bar = 50 µm. (n=5–7 mice per group); (**C**) Quantification of total fibrotic area in cardiac tissue (% of total area); (**D**) Hydroxyproline

*Figure 1 continued on next page*

*Figure 1 continued*

content of mouse heart from young and old wt and *Arg2*[-/-] female mice. (n=4 mice in each group); (**E**) Representative confocal images showing immunofluorescence staining of PDGF-Rα (green, fibroblasts marker) in young and old wt and *Arg2*[-/-] heart tissue. DAPI (blue) is used to stain nuclei. Scale bar = 50 μm; (**F**) Relative PDGF-Rα signal quantification of confocal images (n=4 per each group). (**G**) Representative confocal images showing immunofluorescence staining of p16 (green, senescent marker) in young and old wt and *Arg2*[-/-] heart tissue. DAPI (blue) is used to stain nuclei. Scale bar = 50 μm; (**H**) Percentage of p16[+] nuclei in the four groups (n=4). The values shown are mean ± SD. Data are presented as the fold change to the young-*wt* group, except for panel **D**. *p≤0.05, **p≤0.01, ***p≤0.005, ****p≤0.001 between the indicated groups. wt, wild-type mice; *Arg2*[-/-], *Arg2* gene knockout mice.

The online version of this article includes the following figure supplement(s) for figure 1:

**Figure supplement 1.** *Arg2* and *Arg1* expression in male and female mice.

**Figure supplement 2.** Collagen gene expression in female mice.

*Arg2*[-/-] animals (***Figure 1B and C***). The age-associated increase in cardiac fibrosis and its inhibition by *Arg2*[-/-] are further confirmed by quantitative measurements of collagen content as assayed by determination of hydroxyproline levels in the heart tissues (***Figure 1D***). The increased cardiac fibrosis in aging is, however, associated with decreased mRNA levels of *collagen1α* (*Col3a1*) and *collagen3α* (*Col3al*), the major isoforms of pre-collagen in the heart (***Figure 1—figure supplement 2A and B***), which is a well-known phenomenon in cardiac fibrotic remodeling (***Besse et al., 1994***; ***Horn and Trafford, 2016***). The results demonstrate that age-associated cardiac fibrosis and prevention in *Arg2*[-/-] mice are due to alterations of translational and/or post-translational regulations including collagen synthesis and/or degradation. Furthermore, an increased density of fibroblasts as characterized by positive staining of PDGF-Rα is observed in wt old mice, but not in age-matched *Arg2*[-/-] animals (***Figure 1E*** and ***Figure 1F***). Interestingly, an age-associated increase in p16[ink4] positive senescent cells in wt mouse heart is prevented in the *Arg2*[-/-] animals (***Figure 1G and H***).

Moreover, a significant increase in macrophage marker *Adgre1* (corresponding to F4-80 protein; ***Figure 2A***) and elevated gene expression of numerous pro-inflammatory cytokines such as *Mcp1*, *Il1b*, and *Tnfa* are observed in aging heart of wt mice (***Figure 2B–D***). This age-associated increase in the inflammatory markers and cytokine gene expression (except *Tnfa*) is overall significantly reduced or prevented in *Arg2*[-/-] mice (***Figure 2A–D***). In line with the gene expression of *Adgre1*, immunofluorescence staining reveals an age-associated increase in the numbers of F4-80[+] cells in the wt mouse heart, which is reduced in the age-matched *Arg2*[-/-] animals (***Figure 2E–G***), demonstrating that *Arg2* gene ablation reduces macrophage accumulation in the aging heart. Interestingly, resident macrophages characterized by LYVE1[+]/F4-80[+] cells (***Figure 2E and H***; ***Figure 2—figure supplement 1A***) are predominant in the aging heart as compared to the infiltrated CCR2[+]/F4-80[+] cells (***Figure 2F and I***; ***Figure 2—figure supplement 1B***). The increase in both LYVE1[+]/F4-80[+] and CCR2[+]/F4-80[+] macrophages in aging heart is reduced in *Arg2*[-/-] mice (***Figure 2E, F, H and I***).

We also observed an age-associated increase in total numbers of apoptotic cells as demonstrated by TUNEL staining in the heart of wt mice, which is reduced in age-matched *Arg2*[-/-] animals (***Figure 3A*** and ***Figure 3B***). Further experiments with wheat germ agglutinin (WGA)-Alexa Fluor 488 (used to define cell borders; ***Figure 3C***) show a higher percentage of apoptotic cardiomyocytes (CM) than non-cardiomyocytes (NCM) in wt mice (***Figure 3D***). This increased percentage of cardiomyocyte apoptosis is also prevented in age-matched *Arg2*[-/-] animals (***Figure 3D***). The percentage of apoptotic non-cardiomyocytes in aged *Arg2*[-/-] mice tended to be reduced as compared to wt mice, but it did not reach statistical significance (***Figure 3D***).

In addition, heart from old wt mice reveals elevated levels of SNAIL (the master regulator of End-MT process) and vimentin (mesenchymal marker), which is prevented in *Arg2*[-/-] mice (***Figure 4A*** to ***Figure 4C***). No changes in CD31 (endothelial marker) levels are observed among the four groups (***Figure 4A and D***). Immunofluorescence staining reveals that the age-related up-regulation of vimentin is mostly due to expansion of fibroblast population in left ventricle as shown in ***Figure 1E*** and ***Figure 1F***. Confocal co-immunofluorescence staining shows that vimentin could be found in CD31[+] endothelial cells of blood vessels in the heart of old wt but rarely in age-matched *Arg2*[-/-] mice (***Figure 4E***). The results suggest at least a partial End-MT occurring in cardiac aging that is prevented by *Arg2* ablation. With aging, the heart weight to body weight ratio (HW/BW) increases to a similar extent between wt and *Arg2*[-/-] mice (***Figure 4—figure supplement 1***), suggesting no difference in age-related cardiac hypertrophy between wt and *Arg2*[-/-] mice.

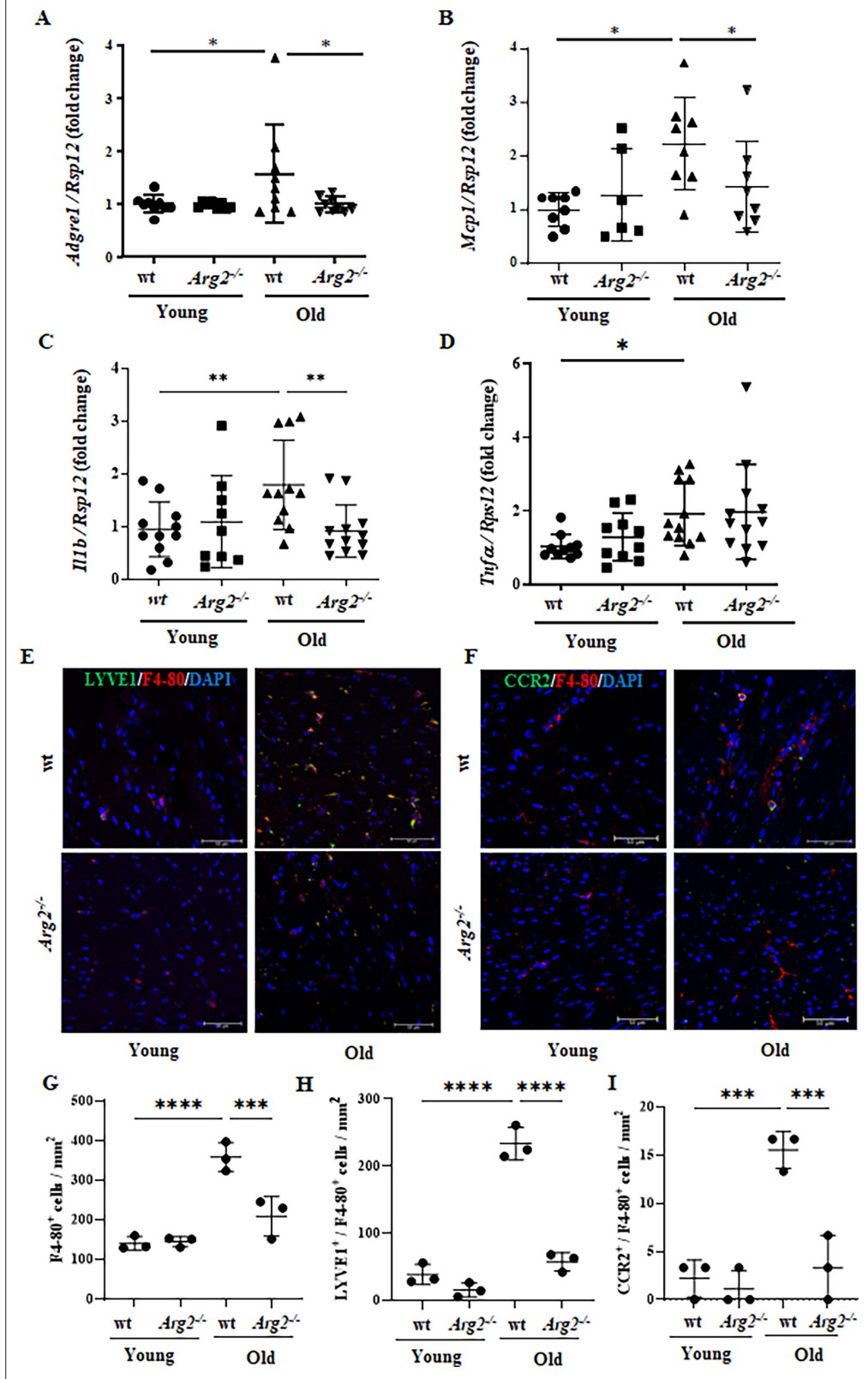

**Figure 2.** Age-associated elevation in inflammatory cytokines and apoptosis in heart is prevented in female *Arg2-/-* mice. mRNA expression levels of (**A**) *Adgre1*, (**B**) *Mcp-1*, (**C**) *Il1b*, and (**D**) *Tnfa* in young and old wt and *Arg2-/-* female mouse hearts were analyzed by qRT-PCR. *Rps12* served as the reference (n=6–12 mice per group); (**E**) Representative confocal images of young and old wt and *Arg2-/-* heart tissue showing co-localization of LYVE1

*Figure 2 continued on next page*

*Figure 2 continued*

(green) and F4-80 (red, mouse macrophage marker), (**F**) Representative co-localization images of CCR2 (green) and F4-80 (red) in wt and *Arg2*⁻/⁻ heart tissues. DAPI (blue) is used to stain nuclei. Scale bar = 50 µm. Graph showing the quantification of (**G**) F4-80⁺ cells, (**H**) LYVE1⁺/F4-80⁺ cells, and (**I**) CCR2⁺/F4-80⁺ cells per mm² in heart tissue of wt and *Arg2*⁻/⁻ young and old mice (n=3 per each group). The values shown are mean ± SD. Data are expressed as fold change to the young wt group, except for panels **G** to **I**. *p≤0.05, **p≤0.01, ***p≤0.005, and ****p≤0.001 between the indicated groups. wt, wild-type mice; *Arg2*⁻/⁻, *Arg2*⁻/⁻ gene knockout mice.

The online version of this article includes the following figure supplement(s) for figure 2:

**Figure supplement 1.** Validation of CCR2 and LYVE1 antibodies.

## Cellular localization of ARG2 in aging heart

Next, the cellular localization of ARG2 in aging heart is investigated. Since *Arg2* mRNA levels are upregulated mostly in aged females, heart tissues from old female mice are used mainly for this purpose. To our surprise, confocal co-immunofluorescence staining does not show ARG2 in cardiomyocytes

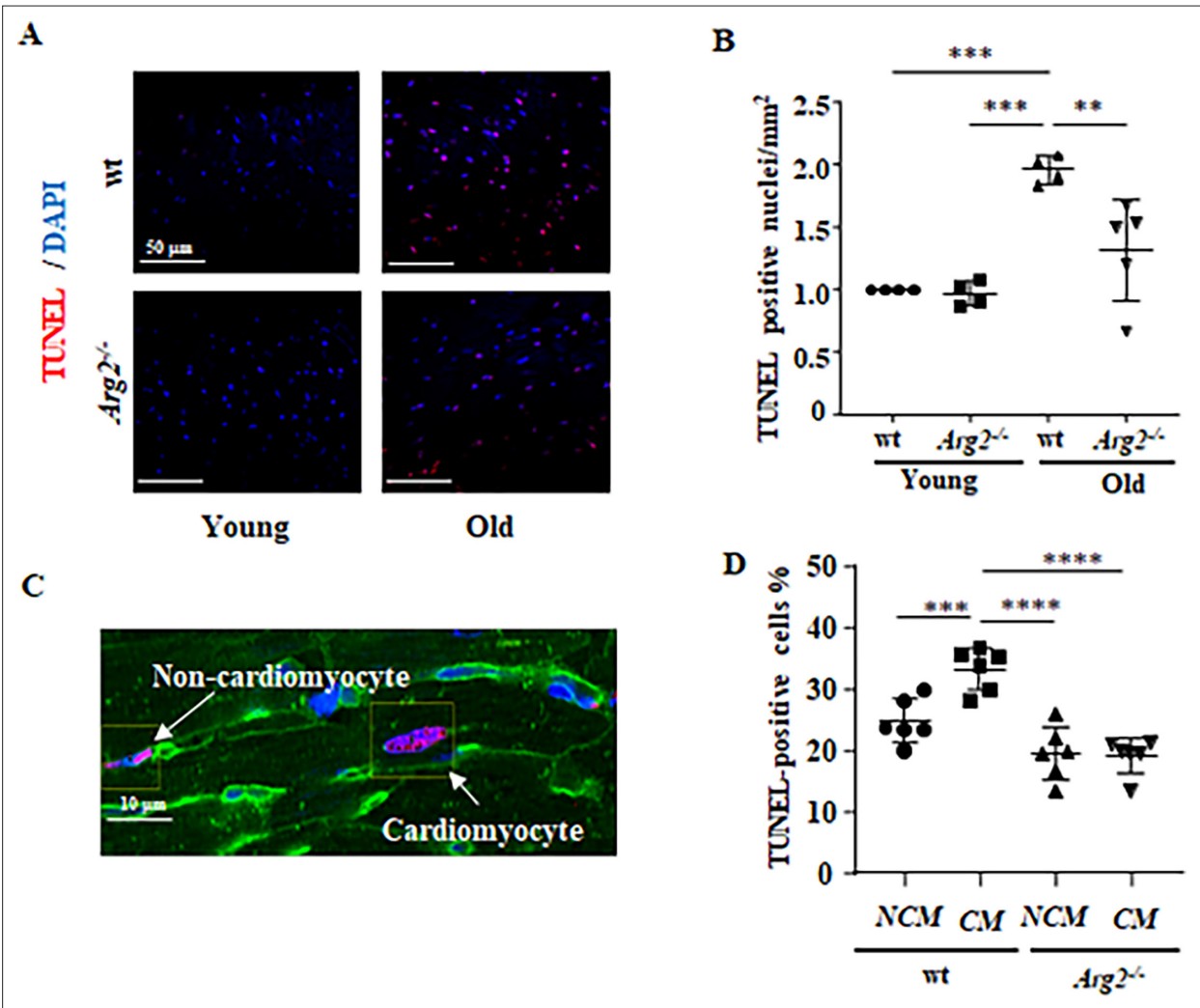

**Figure 3.** Age-associated elevation in apoptotic cardiomyocytes is prevented in female *Arg2*⁻/⁻ mice. (**A**) Representative confocal images and relative quantification of apoptotic cardiac cells in young and old wt and *Arg2*⁻/⁻ heart tissue. DAPI (blue) is used to stain nuclei. Scale bar = 50 µm; (**B**) graph showing the quantification of the TUNEL-positive cells in old wt and *Arg2*⁻/⁻ hearts; (**C**) Wheat germ agglutinin (WGA)-Alexa Fluor 488-conjugate was used to stain cell membrane, and separate cardiomyocytes and non-cardiomyocytes apoptotic cells. Cell distinction was based on cell size and shape. Scale bar = 10 µm; (**D**) Graphs showing the quantification of the TUNEL-positive cardiomyocytes (CM) and non-myocytes (NCM) in old wt and *Arg2*⁻/⁻ hearts. The values shown are mean ± SD. Data are expressed as fold change to the young wt group, except for panel **D**. **p≤0.01, ***p≤0.005 and ****p≤0.001 between the indicated groups. wt, wild-type mice; *Arg2*⁻/⁻, *Arg2* gene knockout mice.

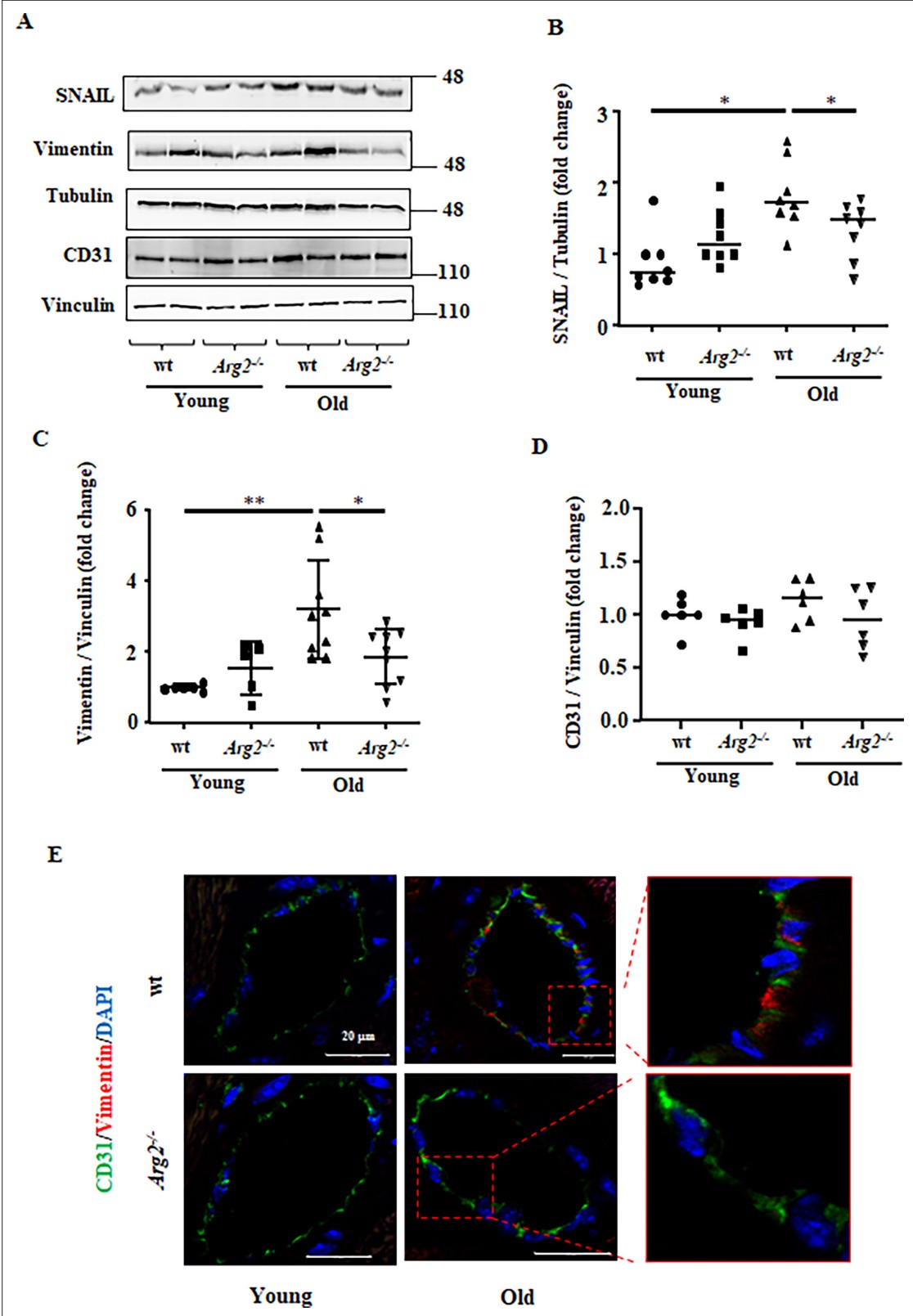

**Figure 4.** Age-related endothelial-to-mesenchymal transition (EndMT) in heart tissue. (**A**) Immunoblotting analysis of CD31 (endothelial marker), vimentin (mesenchymal marker), and SNAIL (master regulator of EndMT) in the heart of wt and *Arg2−/−* female young and old mice; tubulin and vinculin served as protein loading controls. Molecular weight (kDa) is indicated at the side of the blots. The plot graphs show the quantification of the SNAIL (**B**), vimentin (**C**), and CD31 (**D**) signals on immunoblots (n=6–10 mice in each group); (**E**) Representative confocal images showing co-localization

*Figure 4 continued on next page*

*Figure 4 continued*

of CD31 (green) and vimentin (red) in young and old wt and *Arg2*[-/-] heart tissues. DAPI (blue) is used to stain nuclei. Scale bar = 20 μm. *p≤0.05 and **p≤0.01, between the indicated groups. wt, wild-type mice; *Arg2*[-/-], *Arg2* gene knockout mice.

The online version of this article includes the following figure supplement(s) for figure 4:

**Figure supplement 1.** Age-related heart tissue hypertrophy.

as evidenced by lack of ARG2 staining in Troponin T (TNNT)[+]-cells, a specific cardiomyocyte marker (*Figure 5A*). ARG2 is only found in non-cardiomyocytes (*Figure 5A*). In line with this notion, ARG2 is not detectable by immunoblotting in isolated primary cardiomyocytes from old wt mice at baseline or after exposure to hypoxia (1% $O_2$, 24 hr, *Figure 5B*), a well-known strong stimulus for ARG2 upregulation in many cell types (*Liang et al., 2019*; *Liang et al., 2021*; *Ren et al., 2022*; *Zhu et al., 2023*). In contrast to cardiomyocytes, cardiac fibroblasts isolated from old wt mice show ARG2 protein which is up-regulated by hypoxia (*Figure 5B*). The absence of *Arg2* mRNA in cardiomyocytes in contrast to fibroblasts isolated from wt old mouse hearts, is also confirmed with qRT-PCR (*Figure 5C*). The absence of ARG2 in cardiomyocytes is also demonstrated in the heart of other species such as rats, in which no staining could be found in the cardiomyocytes (with or without infarction) but in non-cardiomyocytes that is cells between myocytes and endocardial cells (*Figure 5—figure supplement 1*). The specificity of the antibody against ARG2 was confirmed in the old wt and *Arg2*[-/-] mouse kidneys in which ARG2 is constitutively expressed in the S3 proximal tubular cells (*Figure 5—figure supplement 1A*) as demonstrated by our previous study (*Huang et al., 2016*). Again, ARG2 staining could be only found in cells between cardiomyocytes in the old wt mice but not in *Arg2*[-/-] animals (*Figure 5—figure supplement 1C*). The absence of ARG2 in cardiomyocytes is also confirmed in human heart tissue biopsies (please see the results in *Figure 5H*). Moreover, ARG2 is detected in MAC-2[+] macrophages (*Figure 5D*), CD31[+] endothelial cells (*Figure 5E*), and PDGF-Rα[+]-fibroblasts (*Figure 5F*), but not in α-SMA[+] vascular smooth muscle cells (*Figure 5G*). These results demonstrate that ARG2 in aging mouse heart is mostly expressed in non-cardiomyocytes such as macrophages, endothelial cells, and fibroblasts. Similar to mouse heart, in human heart (obtained from a 66-year-old woman and a 76-year-old man with no significant cardiac pathology), ARG2 is also absent in cardiomyocytes as proved by lack of co-localization with TNNT[+]-cells (*Figure 5H*), and is expressed in CD31[+]-endothelial cells (*Figure 5I*), CD68[+]-macrophages (*Figure 5J*), and vimentin[+]- (*Figure 5K*) or α-SMA[+]- (*Figure 5L*) fibroblasts/myofibroblasts. In contrast to mouse heart, ARG2 could be found in very few vascular smooth muscle cells in human heart (*Figure 5M*).

## Role of ARG2 in crosstalk between macrophages and cardiomyocytes through IL-1β

Given that age-associated cardiomyocyte apoptosis is prevented in *Arg2*[-/-] mice despite absence of ARG2 in the cardiomyocytes, the non-cell-autonomous effects of ARG2 on cardiomyocytes are then investigated. As reported above, macrophage numbers and *Il1b* mRNA expression are increased in aging heart and reduced in *Arg2*[-/-] mice (*Figure 2A, C and G*). Both immunoblotting and immunofluorescence staining confirm that IL-1β protein levels are higher in old wt heart tissues when compared to age-matched *Arg2*[-/-] (*Figure 6A–D*). Furthermore, co-immunofluorescence staining reveals that IL-1β is localized in Mac2[+]- macrophages (*Figure 6E*). Taking into account that our previous studies demonstrated a relationship of ARG2 and IL-1β in vascular disease and obesity (*Ming et al., 2012*) and in age-associated organ fibrosis such as renal and pulmonary fibrosis (*Huang et al., 2021*; *Zhu et al., 2023*), and IL-1β has been shown to play a causal role in patients with coronary atherosclerotic heart disease as shown by CANTOS trials (*Ridker et al., 2017*), we therefore focused on the role of IL-1β in crosstalk between macrophages and cardiac cells such as cardiomyocytes, fibroblasts, and endothelial cells. Considering that the ablation of the *Arg2* gene specifically reduces the number of TUNEL-positive cardiomyocytes (*Figure 3C and D*), we hypothesized that ARG2 in macrophages may prime cardiomyocyte apoptosis. To test this hypothesis, splenic macrophages are isolated from young and old wt and age-matched *Arg2*[-/-] mice, and conditioned medium from the cells is collected. Elevated ARG2 and IL-1β protein levels are observed in old wt macrophages as compared to the cells from young animals (*Figure 7A–C*). In line with this finding, IL-1β levels in conditioned medium from old wt mouse macrophages are higher as compared to the conditioned medium from young macrophages

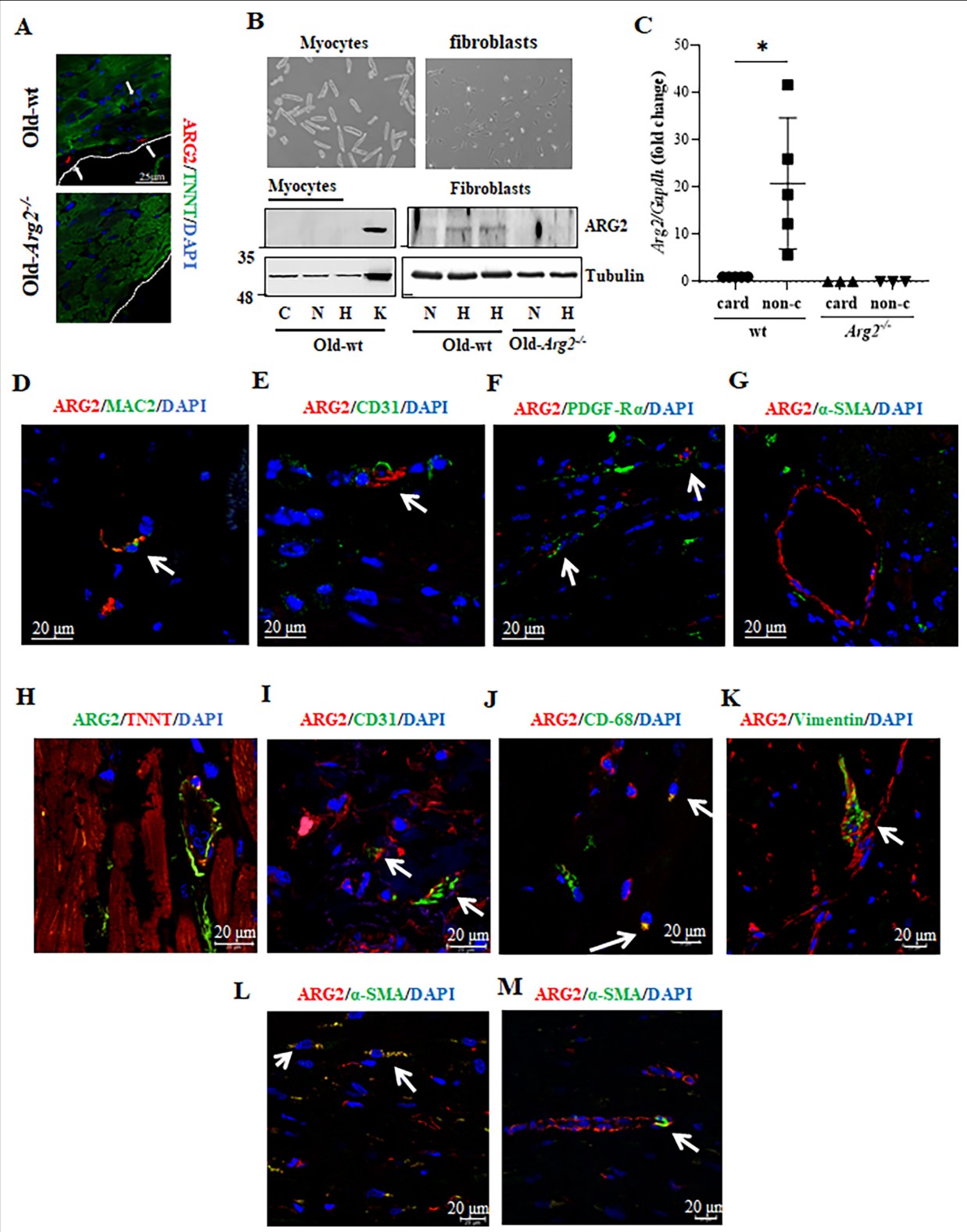

**Figure 5.** Cellular localization of ARG2 in aging heart of female mice. (**A**) Confocal microscopy illustration of immunofluorescence double staining of ARG2 (red) and TNNT (green; cardiomyocytes marker). Scale bar = 25 μm; (**B**) Bright-field microscopy images of isolated wt cardiomyocytes and primary cardiac fibroblasts. The immunoblot shows the level of ARG2 in both cardiomyocytes and fibroblasts upon exposure to hypoxia (1% O₂) for 24 hr. **C** indicates freshly isolated cardiomyocytes used as control, **N** and **H** indicate normoxia and hypoxia conditions, and K indicates kidney tissue extract used as positive control. Tubulin served as protein loading control; (**C**) mRNA expression levels of *Arg2* in cardiomyocytes (card) and non-cardiomyocytes

*Figure 5 continued on next page*

*Figure 5 continued*

(non-c) cells isolated from old wt and *Arg2* $^{-/-}$ female mouse hearts. *Gapdh* served as the reference. (n=3–5 mice per group); (**D** to **G**) Representative confocal images of old wt mouse heart showing co-localization of (**D**) ARG2 (red) and MAC-2 (green, mouse macrophage marker), (**E**) ARG2 (red) and CD31 (green, endothelial marker), (**F**) ARG2 (red) and PDGF-Rα (green, fibroblasts marker), and (**G**) ARG2 and α-smooth muscle actin (α-SMA; green, smooth muscle cell/myofibroblasts marker); (**H** to **M**) Representative confocal images of human heart tissue showing co-localization of (**H**) ARG2 (green) and TNNT (red, cardiomyocytes marker), (**I**) ARG2 (red) and CD31 (green, endothelial marker), (**J**) ARG2 (red) and CD-68 (green, macrophage marker), (**K**) ARG2 (red) and vimentin (green, fibroblast marker), and (**L–M**) ARG2 (red) and α-smooth muscle actin (α-SMA; green), (**L**) myofibroblasts and (**M**) smooth muscle cell marker. DAPI (blue) stains cell nuclei. Scale bar = 20 µm. Each experiment was repeated with 3–5 animals.

The online version of this article includes the following figure supplement(s) for figure 5:

**Figure supplement 1.** ARG2 localization in mouse and rat tissues.

(***Figure 7D***). Interestingly, *Arg2* ablation substantially reduces IL-1β levels in the macrophages (***Figure 7A, C and D***). Importantly, cardiomyocytes isolated from adult (5 months) female wt mice as bioassay cells, when incubated with conditioned medium from old (but not from young) wt macrophages for 24 hr (***Figure 7E***), display more apoptosis as monitored by TUNEL assay (***Figure 7F and G***). This age-associated cardiac damaging effect of macrophages is not observed with *Arg2*$^{-/-}$ macrophage conditioned medium (***Figure 7F and G***), suggesting a role of ARG2-expressing macrophages in mediating cardiomyocyte apoptosis in aging. To further examine whether IL-1β from macrophages is the paracrine mediator for cardiomyocyte apoptosis, the above-described experiment is performed in the presence of the IL-1 receptor antagonist IL-Ra. Indeed, cardiomyocyte apoptosis induced by the conditioned medium from old wt macrophages is abolished by IL-Ra (***Figure 7F and G***).

Similar results are obtained with Raw264.7 cells, a widely used mouse macrophage model. Raw264.7 cells stimulated with lipopolysaccharide (LPS; 100 ng/ml) for 24 hr show a significant increase in both ARG2 and IL-1β, which is inhibited by silencing *Arg2* (***Figure 7—figure supplement 1A to C***), confirming the role of ARG2 in the production of IL-1β in macrophages (***Figure 7—figure supplement 1A and C***). In agreement with the results shown in ***Figure 7***, conditioned medium from LPS-treated Raw264.7 cells significantly enhances cardiomyocyte apoptosis as assessed by TUNEL staining, which is reduced by either *Arg2* gene silencing or by IL-1 receptor blocker ILRa (***Figure 7—figure supplement 1D and E***). It is of note that LPS alone does not induce cardiomyocyte apoptosis at the concentration tested (data not shown), confirming the role of ARG2-mediated IL-1β release from macrophages in cardiomyocyte apoptosis.

Next, we analyzed the relationship between ARG2 and NOS2 in regulation of IL-1β production in macrophages. In the human THP1 monocytes in which ARG2 but not NOS2 is induced by LPS (100 ng/ml for 24 hr; ***Figure 7—figure supplement 2A***), mRNA and protein levels of IL-1β are markedly reduced in *Arg2* knockout THP1 as compared to the wt cells (***Figure 7—figure supplement 2B and C***), further confirming that ARG2 promotes IL-1β production as already shown in RAW264.7 macrophages (***Figure 7—figure supplement 1A and C***). Moreover, in the mouse bone-marrow-derived macrophages, LPS-induced IL-1β production is inhibited by *Nos2* (NOS2 gene) deficiency (*Nos2* knockout BMDM vs wt BMDM; ***Figure 7—figure supplement 2D and E***), while ARG2 levels are slightly enhanced in the *Nos2* knockout BMDM cells (***Figure 7—figure supplement 2D and F***). All together, these results suggest that ARG2 and NOS2 are upregulated by LPS independently, and NOS2 slightly reduces ARG2 protein level. Both ARG2 and NOS2 are required for maximal IL-1β production upon LPS stimulation as illustrated in ***Figure 7—figure supplement 2G***.

## Role of ARG2 in crosstalk between macrophages and cardiac fibroblasts through IL-1β

Since cardiac fibrosis is accompanied by macrophage accumulation in our aging heart model, a crosstalk between macrophages and fibroblasts is then investigated. Cardiac fibroblasts isolated from adult (5 months) female wt mice as bioassay cells produce more hydroxyproline when they are incubated with conditioned medium from old as compared to young wt macrophages (***Figure 8—figure supplement 1A***). This effect of old macrophages on fibroblasts is reduced with conditioned medium from *Arg2*$^{-/-}$ macrophages (***Figure 8A***). In parallel, expression levels of genes involved in extracellular matrix remodeling such as *Fn1* (fibronectin), *Col3a1*, *Tgfb1* (***Figure 8B*** to ***Figure 8D***) as well as *Mmp2* and *Mmp9* (***Figure 8—figure supplement 1B and C***) are reduced in fibroblasts stimulated with conditioned medium from *Arg2*$^{-/-}$ macrophages as compared to the old wt macrophages, while expression

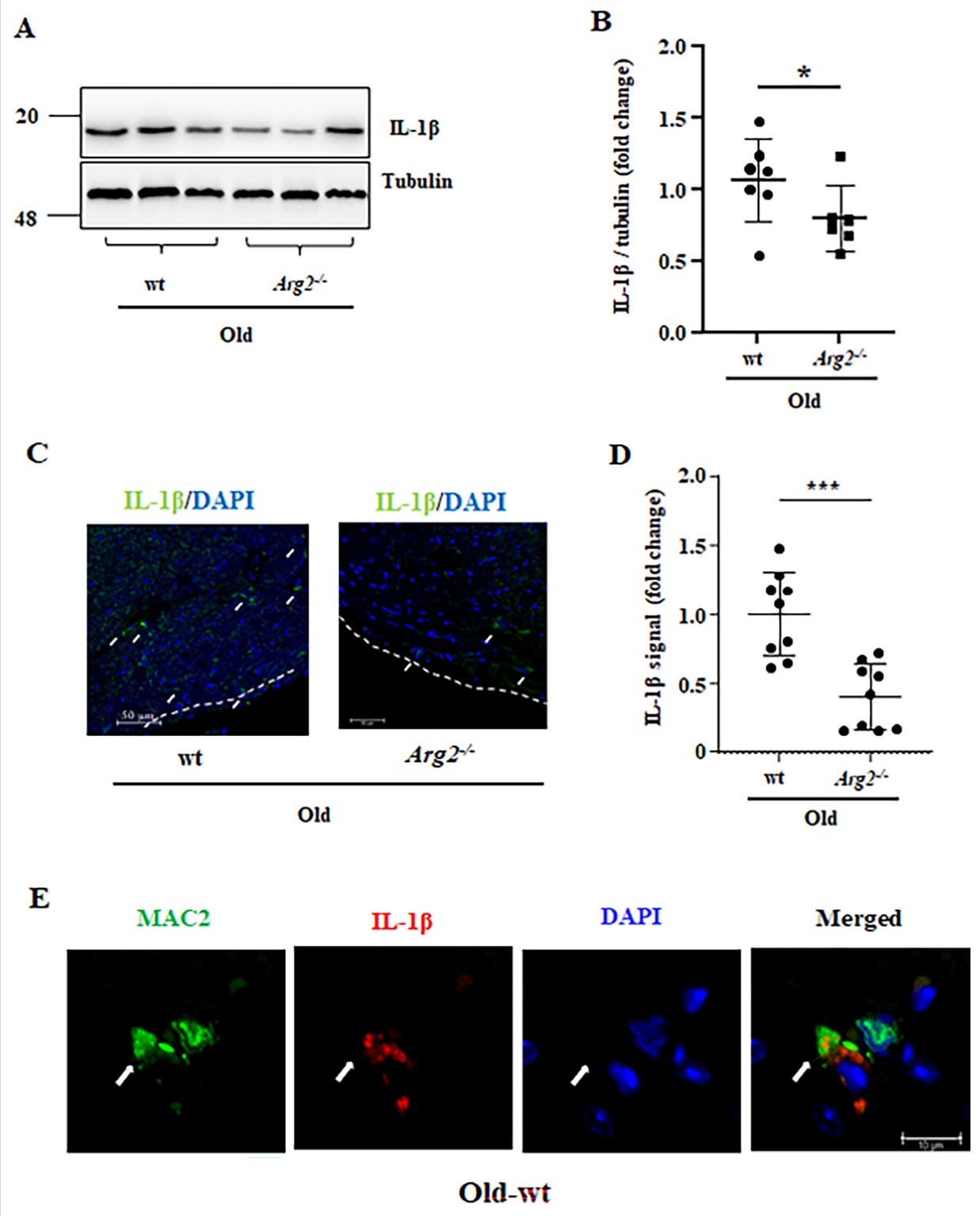

**Figure 6.** ARG2 ablation reduces Il-1β protein levels in aging heart. (**A**) Immunoblotting analysis of Il-1β (active form) in the heart of old wt and *Arg2 -/-* female mice; tubulin served as protein loading control. Molecular weight (kDa) is indicated at the side of the blots; (**B**) The plot graph shows quantification of the Il-1β signals on immunoblots (n=6–7 mice in each group); (**C**) Representative confocal images showing Il-1β localization in old wt and *Arg2 -/-* heart tissues. DAPI (blue) is used to stain nuclei. Scale bar = 50 μm; (**D**) Relative Il-1β signal quantification of confocal images (n=9 per each group); (**E**) Representative confocal images showing co-localization of MAC-2 (green, mouse macrophage marker), and Il-1β (red) in wt heart tissues. This experiment was repeated with 3 animals. Scale bar = 10 μm. *p≤0.05 and ***p≤0.005, between the indicated groups. wt, wild-type mice; *Arg2 -/-*, *Arg2* gene knockout mice.

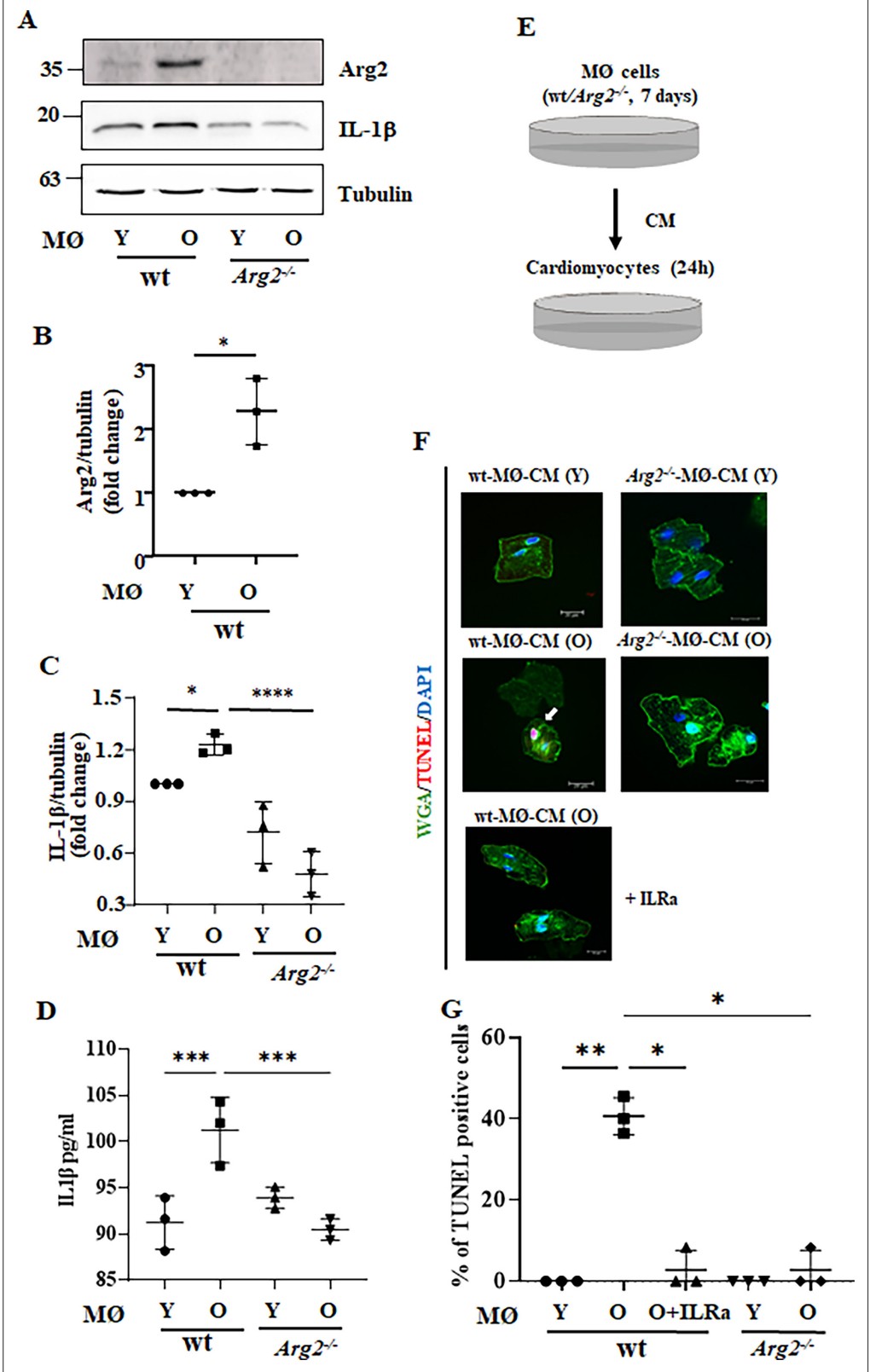

**Figure 7.** In vitro study of crosstalk between aging splenic macrophages and cardiomyocytes. (**A**) Immunoblotting analysis of ARG2 and Il-1β (active form) in mouse splenic macrophages isolated from young and old wt and *Arg2* $^{-/-}$ female mice; tubulin served as protein loading control. Molecular weight (kDa) is indicated at the side of the blots. The plot graphs show the quantification of ARG2 (**B**) and IL-1β (**C**) protein signals on immunoblots

*Figure 7 continued on next page*

*Figure 7 continued*

(n=3 mice in each group); (**D**) IL-1β levels in the conditioned medium from young and old wt and *Arg2* $^{-/-}$ splenic macrophages were measured by ELISA; (**E**) Schematic representation of in vitro crosstalk study; (**F**) Representative confocal images of adult isolated mouse cardiomyocytes stimulated with conditioned media (CM) from young and old, wt and *Arg2* $^{-/-}$ splenic cells (24 hr incubation). Wheat Germ Agglutinin (WGA)-Alexa Fluor 488-conjugate was used to stain cell membrane, and TUNEL was performed to identify apoptotic cells. DAPI is used to stain nuclei. Interleukin receptor antagonist (ILRA; 50 ng/ml) is used to prevent IL-1β binding to its receptor; (**G**) Quantification of TUNEL-positive cardiomyocytes (% in respect to total number of cells). Scale bar = 20 μm. *p≤0.05, **p≤0.01, ***p≤0.005 and ****p≤0.001 between the indicated groups. MØ, splenic macrophage; Y, young; O, old; wt, wild-type mice; *Arg2* $^{-/-}$, *Arg2* gene knockout mice.

The online version of this article includes the following figure supplement(s) for figure 7:

**Figure supplement 1.** In vitro crosstalk between LPS-activated RAW 264.7 macrophages and cardiomyocytes.

**Figure supplement 2.** Effects of ARG2 and NOS2 in regulation of IL-1β production in macrophages.

---

of *Col1a1* is not altered under this condition (*Figure 8—figure supplement 1D*). Importantly, the increase in hydroxyproline levels in cardiac fibroblasts incubated with conditioned medium from old wt macrophages is inhibited by ILRa (*Figure 8E*).

## Cell-autonomous effects of ARG2 in cardiac fibroblasts: role of mtROS

Since ARG2 is expressed in fibroblasts, a cell-autonomous effect of ARG2 in these cells is also investigated in cultured human cardiac fibroblasts (HCF). Overexpression of *Arg2* in the cells (*Figure 8F and G*) increases *Vim* (vimentin) and *Col3a1* expression levels (*Figure 8H and I*), demonstrating a cell-autonomous effect of ARG2 in cardiac fibroblasts. Interestingly, upregulation of *Col3a1* expression by ARG2 is accompanied by increased mitochondrial ROS (mtROS) generation, and inhibition of mtROS by mito-TEMPO (10 μmol/l) prevents *Col3a1* upregulation (*Figure 8J, K and L*), suggesting a role of ARG2-induced mtROS in activation of fibroblasts. The effect of ARG2 in promoting mtROS is also confirmed under hypoxic conditions in which endogenous ARG2 is upregulated accompanied by elevated HIF-1α levels (*Figure 8M and O*). The enhanced mtROS generation under hypoxic conditions is inhibited either by mito-Tempo (*Figure 8N*), the specific inhibitor of mtROS, or by *Arg2* silencing (*Figure 8O and P*).

## Role of ARG2 in crosstalk between macrophages and endothelial cells through IL-1β

Results in *Figure 4* suggest EndMT in aging heart, which is regulated by ARG2. The role of ARG2-expressing macrophages in End-MT of endothelial cells is then investigated in cultured cells. Human endothelial cells exposed to macrophage-derived conditioned medium from old (not young) wt mice for 96 hr have no effect on VE-cadherin but enhance N-cadherin and vimentin (mesenchymal markers), and ARG2 protein levels (*Figure 9A* to *Figure 9E*), suggesting a partial End-MT induced by aged macrophages. Interestingly, these effects of macrophages are prevented in old *Arg2* $^{-/-}$ mice (*Figure 9A–E*) or abolished in the presence of IL-1 receptor antagonist IL-Ra (*Figure 9F–J*). The results suggest a role of IL-1β from old wt macrophages in EndMT process.

## *Arg2* gene ablation reduces ischemic damage in aging heart

To study the specific role of ARG2 in heart ischemia/reperfusion (I/R) injury during aging, ex vivo experiments with Langendorff-perfused heart from old wt and *Arg2* $^{-/-}$ mice were performed followed by cardiac functional analysis. The experimental protocol of global ischemia/reperfusion is shown in *Figure 10A*. Under normoxic baseline conditions, the cardiac functions, i.e., dP/dt$_{max}$. (maximal rate of contraction), dP/dt$_{min}$. (maximal rate of relaxation) and left ventricular developed pressure (LVDP) are comparable between wt and *Arg2* $^{-/-}$ old mice (*Supplementary file 1*). *Arg2* ablation improves cardiac function following I/R injury, i.e., the inotropic contractile function as measured by LVDP and dP/dt$_{max}$, lusitropic dP/dt$_{min}$ (*Figure 10B*). The post I/R recovery is significantly improved in *Arg2* $^{-/-}$ mice when compared to age-matched wt animals (*Figure 10B*). In addition to functional recovery, infarct size after I/R injury was analyzed by triphenyl tetrazolium chloride (TTC) staining in old mice. Under the ischemic/reperfusion condition (20 min ischemia and 30 min reperfusion), no myocardial infarct was detected (*Figure 10—figure supplement 1*), while longer ischemia and reperfusion time (30 min

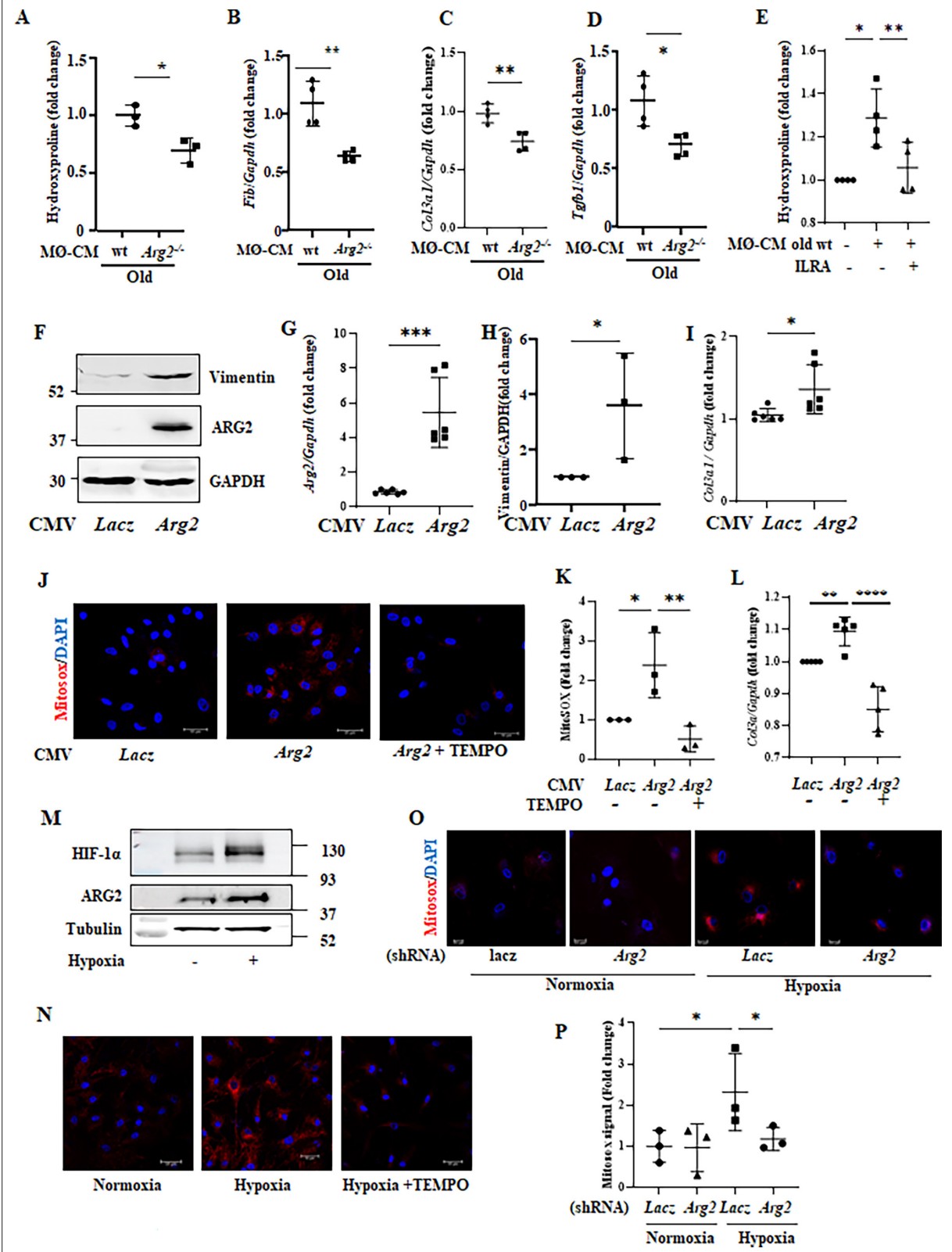

**Figure 8.** Crosstalk between splenic macrophages and cardiac fibroblasts and cell-autonomous effect of ARG2 in cardiac fibroblasts. (**A**) Collagen production measured as hydroxyproline content in mouse wt fibroblasts treated with conditioned media (CM) from old, wt, and *Arg2*⁻/⁻ splenic cells (96 hr incubation) (n=3 mice in each group). The values shown are mean ± SD. mRNA expression levels of (**B**) *Fib*, (**C**) *Col3a1*, and (**D**) *Tgfb1* in wt fibroblasts treated with conditioned media (CM) from old wt and *Arg2*⁻/⁻ splenic cells (96 hr incubation) were analyzed by qRT-PCR. *Gapdh* served as

*Figure 8 continued on next page*

*Figure 8 continued*

the reference. (n=4 mice per group); (**E**) Hydroxyproline content in fibroblasts treated with CM from old wt splenic cells (96 hr incubation). ILRa (50 ng/ml) is used to prevent IL-1β binding to its receptor (n=4 independent experiments). (**F**) Immunoblotting analysis of ARG2 and vimentin in human cardiac fibroblasts (HCFs) upon *Arg2* gene overexpression. GAPDH served as protein loading control; (**G**) qRT-PCR analysis of mRNA expression levels *Arg2* in HCF cells; (**H**) The plot graph shows the quantification of the vimentin signals on immunoblots shown in panel F. (n=3 independent experiments); (**I**) qRT-PCR analysis of mRNA expression levels of *Col3a1* in HCF cells. *Gapdh* served as the reference. (n=6 independent experiments); (**J**) Representative confocal images of human cardiac fibroblasts (HCF) upon transfection with rAd-CMV-Con/ *Arg2* for 48 hr. MitoSOX (Red) is used to stain mitochondrial ROS (mtROS). TEMPO (10 μmol/l) is used to prevent mtROS generation. DAPI (blue) stains cell nuclei. Scale bar = 50 μm. (**K**) Mitosox signal quantification (n=3 independent experiments); (**L**) qRT-PCR analysis of mRNA expression levels of *Col3a1* in HCF cells treated as indicated. *Gapdh* served as the reference. (n=5 independent experiments). (**M**) Immunoblotting analysis of ARG2 and HIF-1α in human cardiac fibroblasts (HCFs) upon 1% hypoxia incubation for 48 h. Tubulin served as a protein loading control; (**N**) Representative confocal images of HCFs under normoxia (21% $O_2$) or hypoxia (1% $O_2$) for 48 hr. MitoSOX (Red) is used to stain mitochondrial ROS (mtROS). TEMPO (10 μmol/l) is used to prevent mtROS generation. DAPI (blue) stains cell nuclei. Scale bar = 50 μm. (**O**) Representative confocal images of HCFs upon transfection with shRNA for *Arg2* gene silencing under normoxia or hypoxia. MitoSOX (Red) is used to stain mitochondrial ROS (mtROS). DAPI (blue) stains cell nuclei. Scale bar = 25 μm. (**P**) MitoSOX signal quantification of images shown in panel O (n=3 independent experiments). Data are expressed as fold change to respective control group. *p≤0.05, **p≤0.01, ***p≤0.005 and ****p≤0.001 between the indicated groups. MØ, splenic macrophage; Con, control.

The online version of this article includes the following figure supplement(s) for figure 8:

**Figure supplement 1.** Crosstalk between splenic macrophages and cardiac fibroblasts.

and 45 min, respectively) is required to induce myocardial infarct, that is 10.5 ± 2.9% in wt old mice, which is significantly reduced in the age-matched *Arg2*[-/-] mice (3.2 ± 0.9%, *Figure 10C and D*). It is to note that following ischemia/reperfusion injury, male mice display cardiac dysfunction, that is reduced left ventricular developed pressure (LVDP), as well as the inotropic and lusitropic states (expressed as dP/dt max and dP/dt min, respectively; *Figure 10—figure supplement 2*). As previously reported (*Murphy and Steenbergen, 2007*), we also found that old male mice are more prone to I/R injury than age-matched female animals. Specifically, 15 min of ischemic time is enough to significantly affect the left ventricle contractile function in the male mice (*Figure 10—figure supplement 2*). As opposed to age-matched old female mice, which are relatively resistant to I/R injury, at least 20 min of ischemia are necessary to induce a significant impairment of the contractile function (*Figure 10*). Similar to females, the post-I/R recovery of cardiac function is also significantly improved in the male *Arg2*[-/-] mice as compared to age-matched wt animals. In addition to functional recovery, triphenyl tetrazolium chloride (TTC) staining (myocardial infarction) upon I/R injury in males is significantly reduced in the age-matched male *Arg2*[-/-] animals (*Figure 10—figure supplement 2C and D*). All together, these results reveal a role for ARG2 in heart function impairment during aging in both genders with a higher vulnerability in males (*Figure 10—figure supplement 2*).

## Discussion

Previous studies have reported the presence of both ARG1 and ARG2 isoenzymes in the cardiac tissues of various animal models including feline, pig, rats, rabbit, and mice (*Gonon et al., 2012*; *Heusch et al., 2010*; *Jung et al., 2006*; *Jung et al., 2010*; *Steppan et al., 2006*). These isoenzymes have been implicated in cardiac ischemic injury and cardiac aging (*Jung et al., 2010*; *Xiong et al., 2017a*). It has been widely assumed that ARG1 and ARG2 are expressed in cardiomyocytes, where they play a role in cardiac injury via a cell-autonomous effect on the cardiomyocytes. However, the cellular localization of arginase isoforms has not been convincingly demonstrated due to the following two reasons: species-specific expression of arginase isoforms and lack of including negative controls with gene knockout samples.

Our current study systematically investigated the expression and cellular localization of the mitochondrial ARG2 in the heart of a natural aging mouse model and in human biopsies. We found an age-associated increase in *Arg2* but not *Arg1* in the heart of male and female mice. Data extracted from a high-throughput sequencing database of male mice (no data on females are available, unfortunately) confirms our findings showing the age-dependent enhancement of *Arg2* but not *Arg1* expression in the heart (GSE201207; *Wolff et al., 2023*; *Figure 1—figure supplement 1C and D*). Surprisingly, we found that neither *Arg2* gene expression nor its protein levels were detectable in cardiomyocytes from either young or old mice. Despite increased *Arg2* mRNA (but not *Arg1* mRNA) expression in the aging heart, mitochondrial ARG2 was absent from cardiomyocytes. This observation aligns with

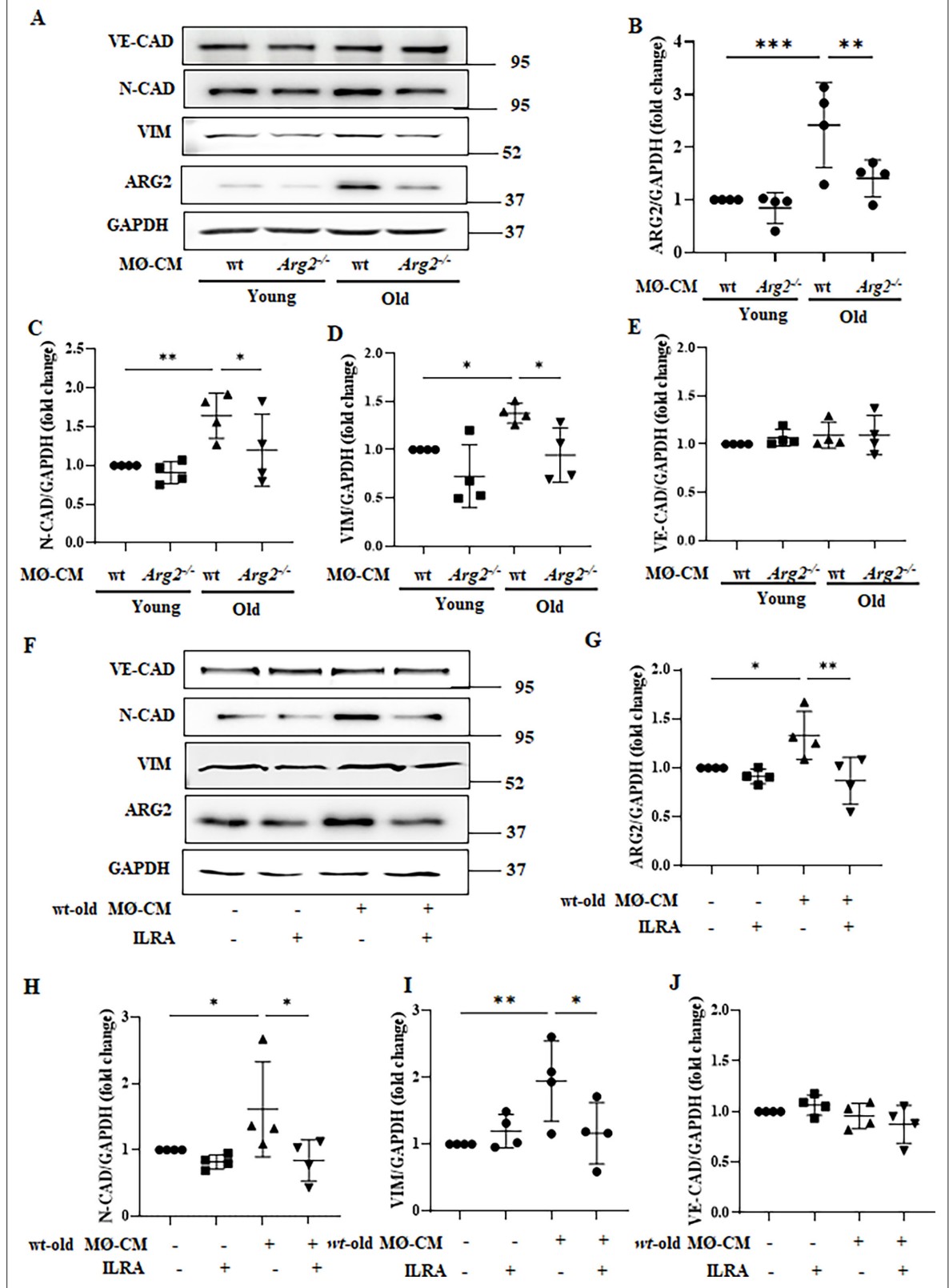

**Figure 9.** Aging wt macrophages induces EndMT in HUVEC. (**A**) Immunoblotting analysis of VE-Cadherin (endothelial marker), N-Cadherin and vimentin (both mesenchymal markers), and ARG2 in HUVEC cells upon incubation with conditioned media (CM) from young and old wt and *Arg2⁻/⁻* splenic cells (4 days incubation); GAPDH served as protein loading control. Molecular weight (kDa) is indicated at the side of the blots. The plot graphs show the quantification of Arg-II (**B**), N-Cadherin (**C**), Vimentin (**D**), and VE-Cadherin (**E**) protein levels (n=3 independent experiments); (**F**) Immunoblotting analysis

*Figure 9 continued on next page*

*Figure 9 continued*

of VE-Cadherin, N-Cadherin, Vimentin, and ARG2 in HUVEC cells upon incubation with CM from old wt splenic cells (4 days incubation); Interleukin receptor antagonist (IL-Ra; 50 ng/ml) is used to prevent IL-1β effect. GAPDH served as protein loading control. Molecular weight (kDa) is indicated at the side of the blots. The plot graphs show the quantification of Arg2 (**G**), N-Cadherin (**H**), Vimentin (**I**), and VE-Cadherin (**J**) protein signals on immunoblots (n=4 independent experiments); Data are expressed as fold change to respective control group. *p≤0.05, **p≤0.01 and ***p≤0.005 between the indicated groups. MØ, splenic macrophage; wt, wild-type mice; *Arg2⁻ᐟ⁻*, *Arg2* gene knockout mice.

previous findings in vascular endothelial cells from mice and humans, where ARG2 (not ARG1) is the predominant isoenzyme (*Ming et al., 2004*). It is to note that a previous study reported that ARG2 is exclusively expressed in isolated cardiomyocytes from rats (*Steppan et al., 2006*). However, we could not confirm this finding. The reason for this discrepancy is not clear. However, we made a great effort to confirm the cellular localization of ARG2 in the heart from different species including mice, rats, and humans. Importantly, negative controls, that is *Arg2⁻ᐟ⁻* samples, are always included in our present study to avoid any possible background signals. Our experiments confirm that ARG2 could be found only in non-cardiomyocytes but not in cardiomyocytes from different species. The experiments with freshly isolated primary cells also confirm that even under hypoxia, a well-known strong stimulus for ARG2 protein levels in different cell types (*Liang et al., 2019*; *Zhu et al., 2023*), no ARG2 protein expression could be detected in the cardiomyocytes but elevated in fibroblasts from old female mice. Furthermore, RT-qPCR could not detect *Arg2* mRNA in cardiomyocytes but in non-cardiomyocytes. All together, these results demonstrate that ARG2 is not expressed or at negligible levels in cardiomyocytes but expressed in non-cardiomyocytes. Further experiments identified that ARG2 is expressed in macrophages, fibroblasts, and endothelial cells in aged mouse heart and human cardiac biopsies. These results suggest that ARG2 exerts non-cell-autonomous effects on aging cardiomyocytes.

It is well known that the aging heart is associated with accumulation of senescent cells, increased cardiomyocyte apoptosis, chronic inflammation, oxidative stress, and cardiac fibrosis (*Vakka et al., 2023*). In the current study, we observe that *Arg2⁻ᐟ⁻* mice are protected from cardiac aging phenotype and reveal a decrease in apoptotic cells, particularly cardiomyocytes but also non-cardiomyocytes. Since cardiomyocytes do not express ARG2 in aging and ARG2 could not even be induced in these cells under hypoxic condition, a non-cell-autonomous effect of ARG2 on cardiomyocyte apoptosis must be involved, meaning that the increased cardiomyocyte apoptosis in aging must be due to paracrine effects of other cell types such as macrophages in which ARG2 levels are elevated in aging. It is evident that crosstalk between cardiomyocytes and non-cardiomyocytes is critical in cardiac functional changes and remodeling (*Hulsmans et al., 2016*). Substantial evidence demonstrates that immune cells, particularly monocytes and/or macrophages, are important contributors to cardiac remodeling in aging and disease conditions (*Hulsmans et al., 2016*; *Jimenez and Lavine, 2022*; *Wang et al., 2020*). Under acute and chronic ischemic heart disease conditions, blood monocytes can be mobilized from bone marrow and spleen and recruited into the injury site of the heart where they play an important role in cardiac remodeling (*Honold and Nahrendorf, 2018*; *Swirski et al., 2009*). An increased accumulation of myeloid cells, including in aging hearts, has been reported and contributes to cardiac inflammaging (*Esfahani et al., 2021*). Besides infiltrated macrophages, tissue resident macrophages also contribute to cardiac inflammation and remodeling in heart disease (*Suku et al., 2022*; *Weinberger et al., 2024*). Interestingly, we demonstrate that total numbers of macrophages in the heart are significantly increased in aging heart, which is due to both enhanced accumulation of resident tissue macrophages and infiltrated macrophages, and both macrophage populations are reduced in *Arg2⁻ᐟ⁻* mice. The enhanced infiltrated macrophages must be due to sustained cardiac injury occurring with aging in the heart accompanied by repairing process, which is manifested by resident macrophage proliferation and fibroblast activation (*Weinberger et al., 2024*). Our previous study demonstrated that ARG2 plays a role in macrophage infiltration in different organ/tissues in high-fat diet and high-cholesterol diet-induced atherosclerosis and obesity mouse models (*Ming et al., 2012*). It remains to be investigated whether ARG2 regulates resident macrophage proliferation in the aging heart and what the underlying mechanisms. Whether infiltrated macrophages can be switched to resident ones and regulated by ARG2 is also an interesting question to be investigated.

As shown above, an increased macrophage accumulation in aging heart associates with elevated expression of numerous inflammatory cytokines in macrophages. This age-associated cardiac inflammation is inhibited in *Arg2⁻ᐟ⁻* animals. Among these inflammatory cytokines, an increase in IL-1β levels

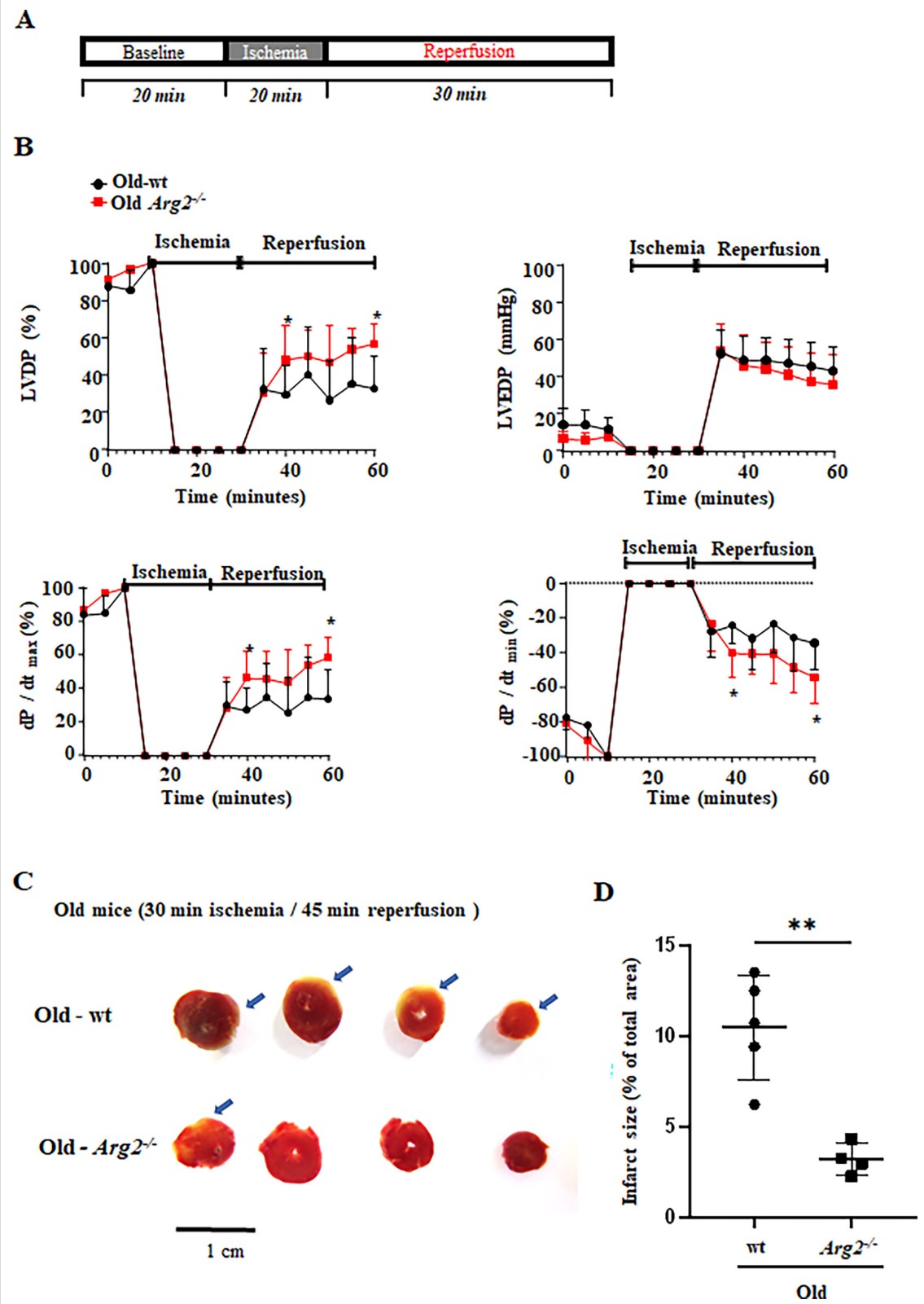

**Figure 10.** ARG2 ablation improves heart function recovery from global ischemia/reperfusion injury (I/R-I). (**A**) Protocol applied for ex vivo Langendorff-heart functional assessment of old wt and age-matched *Arg2⁻/⁻* female mice. After baseline recordings, 20 min global ischemia is followed by 30 min of reperfusion; (**B**) Ex vivo Langendorff-heart assessment of old female wt (black line) and *Arg2⁻/⁻* (red line) hearts. The graphs show the functional recovery of the left ventricular developed pressure (LVDP), left ventricular end diastolic pressure (LVED), and maximal rate of contraction (dP/dt$_{max}$) and

*Figure 10 continued on next page*

**Figure 10 continued**

relaxation (dP/dt$_{min}$). The data are expressed as % of recovery in respect to baseline values and represent the mean ± SD of data from 5 mice per group; (**C**) Representative sections of wt and *Arg2$^{-/-}$* female hearts stained with 2,3,5-TTC. The healthy/normal tissue appears deep red. The arrows indicate the white infarct tissue showing the absence of living cells. (**D**) Relative quantification of infarcted areas. *p≤0.05 and **p≤0.01 between the indicated groups. wt, wild-type mice; *Arg2$^{-/-}$*, *Arg2* gene knockout mice.

The online version of this article includes the following figure supplement(s) for figure 10:

**Figure supplement 1.** Short ischemia/reperfusion induces negligible myocardial infarct in female mice.

**Figure supplement 2.** *Arg2* ablation improves heart function recovery from global ischemia/reperfusion injury (I/R-I) in male mice.

in cardiac tissue and macrophages or monocytes/macrophages derived from spleen of old mice is demonstrated accompanied by elevated ARG2 levels as compared to young animals. Importantly, we showed that ARG2 in the aged macrophages enhances IL-1β expression which mediates cardiomyocyte apoptosis. In support of this notion, cardiomyocyte apoptosis caused by aged macrophages is inhibited in *Arg2$^{-/-}$* mice and prevented by IL-1β receptor antagonist. These results further confirm our previous findings demonstrating a pro-inflammatory role of ARG2 in macrophages in vascular atherosclerosis and high-fat-diet-induced type 2 diabetes mouse models (*Ming et al., 2012*). This conclusion is also further supported by the findings with a mouse macrophage cell line RAW264.7. The question of what the underlying mechanisms are that upregulate ARG2 levels in macrophages in aging remains to be investigated. Moreover, we also found increased apoptosis of other cell types in aging hearts, which is also prevented in *Arg2$^{-/-}$* mice. What are these cells, how is this cell apoptosis triggered, and what is the impact on age-associated cardiac vulnerability to stressors that requires further investigation. Moreover, research has provided evidence that cellular senescence, including , endothelial cells, fibroblasts, and immune cells, all contributes to the cardiac aging and age-associated cardiac dysfunction (*Luan et al., 2024*). Future research shall investigate which cell types are protected from *arg-ii* deficiency from senescence and what the underlying mechanisms are.

One of the interesting questions is the relationship between ARG2 and NOS2 in regulation of IL-1β production in macrophages. Both NOS2 and ARG2 (not ARG1) are highly induced in macrophages in response to several pro-inflammatory stimuli such as LPS, TNFα, and IFN-γ, etc. (*Bogdan, 2015*; *Ming et al., 2012*) and considered as pro-inflammatory M1-macrophage markers (*Martinez and Gordon, 2014*; *Ming et al., 2012*). Since arginase and NOS2 share the same metabolic substrate L-arginine, an increase in arginase levels is proposed to decrease intracellular L-arginine bioavailability for NOS2, and in turn, to reduce pro-inflammatory responses of macrophages (*Chang et al., 1998*). Hence, arginase including the two isoforms ARG1 and ARG2 is regarded as anti-inflammatory (*Yang and Ming, 2014*). ARG1 is indeed associated with M2-anti-inflammatory macrophages (*Yang and Ming, 2014*), while ARG2 (but not ARG1) along with NOS2 is enhanced in pro-inflammatory macrophages upon LPS stimulation as shown in the previous study (*Ming et al., 2012*) and also confirmed in our present study. Importantly, knockout or knockdown of the *Arg2* gene reduces IL-1β production as shown in our present study and decreases pro-inflammatory cytokine levels in vitro and in vivo in various chronic inflammatory disease models and inflammaging (*Atawia et al., 2019*; *Ming et al., 2012*; *You et al., 2013*). These results demonstrate that ARG2, in contrast to ARG1, exerts pro-inflammatory functions in macrophages. Despite controversies published in the literature (*Bogdan, 2015*), NOS2 is generally considered to exert pro-inflammatory effects in macrophages (*Eslami et al., 2017*; *Yao et al., 2022*). This is confirmed by our present study showing that LPS-stimulated IL-1β production in murine macrophages is inhibited when the NOS2 gene *Nos2* is ablated. Since ARG2 promotes IL-1β production and ARG2 levels are increased under *Nos2*-deficiency condition, one would expect an increased IL-1β production in *Nos2* knockout (*Nos2$^{-/-}$*) macrophages. This is, however, not the case. A strong inhibition of IL-1β production in *Nos2$^{-/-}$* macrophages is observed. These results implicate that NOS2 promotes IL-1β production independently of ARG2, and the inhibiting effect of IL-1β by *Nos2* deficiency is dominant and able to counteract ARG2's stimulating effect on IL-1β production. On the other hand, the question of whether ARG2 promotes IL-1β production through NOS2 could also be excluded due to the following reasons. First, in the human THP1 cells in which ARG2 but not NOS2 is induced by LPS, ablation of the *Arg2* gene reduces IL-1β mRNA and protein levels; second, both ARG1 and ARG2 are believed to inhibit endothelial NOS-mediated NO synthesis, *Arg2* deficiency would increase L-arginine bioavailability for NOS activity (*Durante et al., 2007*). Since NOS2 induction by LPS is promoting IL-1β production as demonstrated in our current study, an increase in

IL-1β production by *Arg2* deficiency would be expected under this condition. We observe, however, a decreased IL-1β production in *Arg2*^-/- cells. Hence, our results demonstrate that ARG2 promotes IL-1β production in macrophages independently of NOS2 (This concept is illustrated in the ***Figure 7—figure supplement 2G***). Future study shall further confirm the NOS2-independent stimulating effect on IL-1β production by ARG2 in the *Nos2*^-/- mouse model. It remains to be investigated what the exact molecular mechanisms of IL-1β production promoted by ARG2 and NOS2 in macrophages are, respectively. Since IL-1β is produced and released via activation of inflammasome (***Agostini et al., 2004***; ***Cullen et al., 2015***), whether ARG2 and NOS2 promote macrophage pro-inflammatory responses and IL-1β release through inflammasome activation remains to be studied.

Previous studies suggest that ARG2 exerts both L-arginine metabolizing-dependent and -independent (pleiotropic) effects depending on cell types. For example, in vascular smooth muscle cells in which no NOS exists, even inactive mutant of ARG2 is capable of inducing cell senescence (***Xiong et al., 2013***), while in endothelial cells, eNOS-uncoupling, that is decreased NO and increased superoxide anion production, is dependent on the L-arginine metabolizing activity of ARG2 (***Yepuri et al., 2012***). It is understandable that endothelial dysfunction caused by enhanced ARG2 levels would reduce coronary perfusion and in turn decrease cardiac contractile function as reported in the literature (***Khan et al., 2012***; ***Luo et al., 2014***). Whether ARG2 promotes inflammatory responses in macrophages through the pleiotropic effects remains to be investigated.

It is of note that a study reported that ARG2 is required for IL-10-mediated inhibition of IL-1β in mouse BMDM upon LPS stimulation (***Dowling et al., 2021***), which suggests an anti-inflammatory function of ARG2. The results of our study, however, demonstrate a pro-inflammatory effect of ARG2 in macrophages. Our findings are supported by the study from another group, which shows decreased pro-inflammatory cytokine production including IL-6 and IL-1β in *Arg2*^-/- BMDM most likely through suppression of NFκB pathway, since *Arg2*^-/- BMDM reveals decreased activation of NFκB and IL-1β levels upon LPS stimulation (***Uchida et al., 2023***). Most importantly, our previous study also showed that reintroducing the *Arg2* gene back to the *Arg2*^-/- macrophages markedly enhances LPS-stimulated pro-inflammatory cytokine production (***Ming et al., 2012***), providing further evidence for a pro-inflammatory role of *Arg2* under LPS stimulation. In support of this conclusion, chronic inflammatory diseases such as atherosclerosis and type 2 diabetes (***Ming et al., 2012***), inflammaging in lung (***Zhu et al., 2023***), kidney (***Huang et al., 2021***), and pancreas (***Xiong et al., 2017b***) of aged animals or acute organ injury such as acute ischemic/reperfusion or cisplatin-induced renal injury are reduced in the *Arg2*^-/- mice (***Uchida et al., 2023***). The discrepant findings between these studies and those with IL-10 may implicate dichotomous functions of ARG2 in macrophages, depending on the experimental context or conditions. Nevertheless, our results strongly implicate a pro-inflammatory role of ARG2 in macrophages in the inflammaging in aging heart.

One of the important features of cardiac aging is fibrosis, which is due to chronic inflammation processes (***Lu et al., 2017***). There is convincing evidence demonstrating that chronic inflammation facilitates development of myocardial fibrosis in heart failure which is associated with aging (***Lillo et al., 2023***). A crosstalk between macrophages and fibroblasts has been demonstrated to be important for cardiac fibrosis in diseased conditions (***Hartupee and Mann, 2016***). Studies show that monocyte-derived macrophages from bone marrow and spleen are recruited in advanced age, contributing to cardiac fibrosis and myocardial dysfunction (***Hulsmans et al., 2018***). In our present study using an in vitro cellular model, we show a pivotal role of Arg-II-expressing macrophages in activation of cardiac fibroblasts in aging heart. First, ARG2 protein is present in the cardiac macrophages and enhanced in monocyte-derived macrophages in aging mice; second, conditioned medium from aged *Arg2*^-/- macrophages exhibits decreased potential of stimulating cardiac fibroblasts to produce components that are essentially involved in fibrosis such as hydroxyproline, *Fn1*, *Col3a1*, and *Tgfb1*. Interestingly, the enhanced hydroxyproline levels stimulated by old wt mouse macrophage-derived conditioned medium are inhibited by the IL-1β receptor blocker, demonstrating a paracrine effect of IL-1β release from aged macrophages on cardiac fibroblast activation. Moreover, the age-associated cardiac fibrosis and increase in fibroblast number in the heart is prevented in *Arg2*^-/- mice, demonstrating a crosstalk between macrophage and fibroblasts in cardiac aging.

The fact that cardiac fibroblasts in aging hearts express elevated ARG2 indicates a contribution of a cell-autonomous effect of ARG2 in this cell type to cardiac fibrosis in aging. This concept is confirmed by the experiments showing that overexpression of *Arg2* in cardiac fibroblasts enhances vimentin

levels associated with increased expression of *Col3a1* and mitochondrial ROS generation. Importantly, the inhibition of mitochondrial ROS by TEMPO abolished the effect of *Arg2* overexpression in the fibroblasts, demonstrating that ARG2 exerts its cell-autonomous stimulating effects in fibroblasts through mitochondrial ROS. The role of ARG2 in mitochondrial ROS generation is further confirmed by the results showing that silencing *Arg2* abolished miROS generation in fibroblasts when ARG2 is upregulated under hypoxic conditions. These findings are supported by the studies showing that ROS favors cardiac fibroblast proliferation and activation to produce collagen (*Alili et al., 2014*; *Ohtsu et al., 2005*). With these results, we demonstrate that cardiac fibroblasts are activated in aging by a paracrine effect of macrophages, that is mediated by ARG2/IL-1β and also by a cell-autonomous effect of ARG2 through mtROS.

Endothelial cells play a key role in regulation of blood flow to organs through eNOS pathway, one of the most important regulatory mechanisms in cardiovascular system (*Godo and Shimokawa, 2017*). Previous studies demonstrated a causal role of ARG2 in promotion of vascular endothelial aging through eNOS uncoupling (*Yepuri et al., 2012*). In line with this, our current study showed that ARG2 is expressed in the coronary vascular endothelium of old mice and also in human heart. Emerging evidence also suggests that endothelial cells have a significant capacity for plasticity and can change their phenotype, including endothelial-to-mesenchymal transition (EndMT), a process that endothelial cells undergo extensive alterations, including changes in gene and protein expression, loss of the endothelial phenotype, and switch to mesenchymal phenotype such as increased cell migration, proliferation, and increased collagen production, contributing to cardiovascular disease and fibrosis (*Murdoch et al., 2014*; *Zeisberg et al., 2007*; *Zhang et al., 2021*). Although EndMT in cardiac fibrotic models, particularly in pro-atherosclerotic mouse models, is at least partially evidenced, it is less known in cardiovascular aging (*Evrard et al., 2016*; *Kidder et al., 2023*; *Xu and Kovacic, 2023*). When compared to young animals, aged wt heart tissues show enhanced levels of SNAIL, the master regulator of End-MT process. Moreover, immunofluorescence reveals that CD31⁺ endothelial cells also co-express vimentin (mesenchymal marker). No changes in endothelial marker expression such as CD31 and vascular-endothelial cadherin (data not shown) are appreciated, suggesting that partial End-MT occurs during aging. Ablation of ARG2 blunts the age-related elevation of both vimentin and SNAIL, thereby reducing the number of CD31⁺/vimentin⁺ endothelial cells. This data hints that lower cardiac tissue fibrosis in *Arg2⁻/⁻* is at least partially contributed to by dampened End-MT process. Indeed, the conditioned medium from old wt macrophages can enhance mesenchymal markers N-cadherin and vimentin, as well as ARG2 protein levels in endothelial cells without changes in endothelial markers such as VE-cadherin, confirming a partial End-MT induced by aged macrophages. Interestingly, this effect of macrophages is prevented when *Arg2* was ablated. Furthermore, the paracrine effects of the old macrophages on EndMT are prevented by IL-1 receptor antagonist ILRa, demonstrating a role of IL1β from old wt macrophages expressing ARG2 in EndMT process. Moreover, ARG2 is expressed and upregulated in endothelial cells with aging (*Yepuri et al., 2012*). Overexpression of *Arg2* in the endothelial cells did not affect the End-MT markers (data not shown), suggesting that ARG2 alone is not sufficient to induce a cell-autonomous effect in End-MT. Finally, we demonstrate that *Arg2⁻/⁻* mice in aging exhibit improved cardiac function and are more resistant to ischemic/reperfusion stress and reveal a better recovery and less myocardial infarct area as compared to the old wt animals. This protection against age-related cardiac dysfunction and vulnerability to ischemic stress in *Arg2⁻/⁻* animals could be explained by the above-described cardiac aging phenotype such as increase in apoptosis of cardiac cells, decrease in macrophage-mediated inflammation, endothelial dysfunction and EndMT, and fibroblast activation.

Previous studies provided sex-related differences in ARG2 levels at both mRNA (*Xiong et al., 2017a*) and protein levels in various organs (*Huang et al., 2021*; *Xiong et al., 2017b*). We further confirmed the observation that in female mice, there is a greater age-associated upregulation of *Arg2* levels in the heart, which is specifically found in non-cardiomyocytes but not in cardiomyocytes. A possible explanation for this sex-specific phenomenon may lie in differences in hormonal patterns between male and female animals throughout the aging process. Indeed, sex hormones, including estrogen and testosterone, have been shown to regulate *arg-ii* expression in various tissues. Estrogen seems to downregulate ARG2 (*Hayashi et al., 2006*). The age-related greater increase in *Arg2* in the female mice might be due to loss or lack of estrogen in old female animals, which requires further investigation. However, testosterone has been reported to upregulate *Arg2* expression (*Levillain*

*et al., 2005*). This observation, however, does not seem to support the hypothesis that increased *Arg2* expression in aged male mouse heart is related to sex hormones. Nevertheless, how age enhances ARG2 in a sex-specific manner remains a topic of further investigation. Of note, previous studies showed that sex-related differences in pathological phenotypes in the aging kidney, pancreas, and lung are associated with higher levels of ARG2 in the females (*Huang et al., 2021*; *Xiong et al., 2017b*; *Zhu et al., 2023*). The fact that aged females have higher ARG2 but are more resistant to I/R injury seems contradictory to the detrimental effect of Arg-II in I/R injury. It is presumable that cardiac vulnerability to injuries stressors depends on multiple factors/mechanisms in aging. Other factors/mechanisms associated with sex may prevail and determine the higher sensitivity of male heart to I/R injury, which requires further investigation. Nevertheless, the results of our study show that ARG2 plays a role in cardiac I/R injury also in males.

In conclusion, our study demonstrates that age-related upregulation of ARG2 in macrophages, fibroblasts, and endothelial cells contributes to the cardiac aging phenotype. This includes manifestations, such as cardiac inflammatory response, partial EndMT, fibroblast activation, resulting in cardiomyocyte apoptosis, cardiac fibrosis, and ultimately, myocardial dysfunction. Moreover, it enhances the vulnerability of aging heart to ischemic/reperfusion injury, primarily through a non-cell-autonomous paracrine release of IL-1β from macrophages acting on cardiomyocytes and endothelial cells. This effect is in addition to a cell-autonomous effect of ARG2 in cardiac fibroblasts (*Figure 11*). Future research shall develop the macrophage-specific *Arg2*$^{-/-}$ mouse model to confirm this conclusion with aging animals. Since ARG2 is also expressed in fibroblasts and endothelial cells and exerts cell-autonomous and paracrine functions, aging mouse models with conditional *Arg2* knockout in the specific cell types would be the next step to elucidate cell-specific function of ARG2 in cardiac aging. In this context, another interesting aspect is the crosstalk between macrophages and vascular SMC in the aging heart. In our present study, we could not detect ARG2 in vascular SMC of mouse heart but in that of human heart. This could be due to the difference in species-specific ARG2 expression in the heart or related to the disease conditions in human heart which is harvested from patients with cardiovascular diseases. Indeed, in the *Apoe*$^{-/-}$ mouse atherosclerosis model, aortic SMCs do express Arg-II (*Xiong et al., 2013*). It is interesting to note that rodents hardly develop atherosclerosis as compared to humans. Whether this could be partly contributed to the different expression of ARG2 in vascular SMC between rodents and humans requires further investigation. In our present study, the aspect of the crosstalk between macrophages and vascular SMC is not studied. Since the crosstalk between macrophages and vascular SMC has been implicated in the context of atherogenesis as reviewed (*Gong et al., 2025*), further work shall investigate whether ARG2-expressing macrophages could interact with vascular SMC in the coronary arteries in the heart and contribute to the development of coronary artery disease and/or vascular remodeling and the underlying mechanisms. Nevertheless, our study strongly suggests that targeting ARG2 has a wide spectrum of direct and indirect effects on multiple cell types in aging heart including macrophages, fibroblasts, endothelial cells, and cardiomyocytes, resulting in improved cardiac resistance to ischemia/reperfusion injury, which is due to inhibition of inflammation, fibrosis, and cell apoptosis. Thus, ARG2 shall represent a promising therapeutic target in the treatment of age-related cardiac dysfunction and protect the heart under diseased conditions.

## Materials and methods

### Arg2$^{-/-}$ mouse and sample preparation

Wild type (wt) and *Arg2* knockout (*Arg2*$^{-/-}$) mice were kindly provided by Dr. William O'Brien (*Shi et al., 2001*) and back crossed to C57BL/6 J for more than 10 generations. Genotypes of mice were confirmed by polymerase chain reaction (PCR) as previously described (*Ming et al., 2012*). Offspring of wt and *Arg2*$^{-/-}$ mice were generated by interbreeding from hetero/hetero cross. Mice were housed at 23°C with a 12-hr-light-dark cycle. Animals were fed a normal chow diet and had free access to water and food. The animals were selected and included in the study according to age, gender, and genotype. Only healthy animals, based on scoring of activity, posture, general appearance (coat, skin), locomotion, eyes/nose, body weight loss, were used. In this study, confounders such as the animal/cage location were controlled. All experimenters were aware of the group allocation at the different stages of the experiments. Male and female mice at the age of 3–4 months (young) or 20–22 months

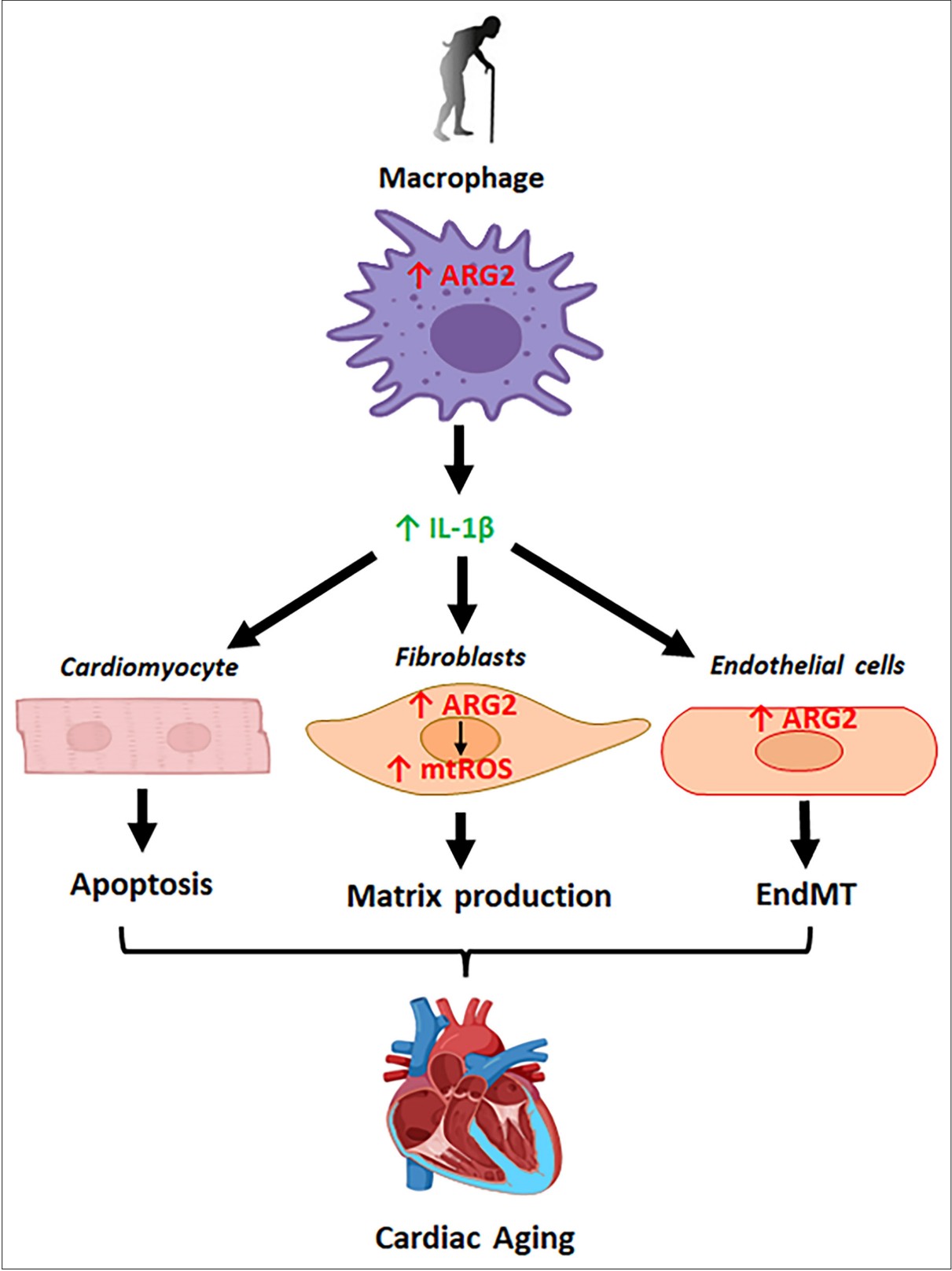

**Figure 11.** Schematic illustration of the role of Arg-II and underlying mechanisms in cardiac aging.

(old) were euthanized under deep anesthesia (i.p. injection of a mixture of ketamine/xylazine 50 mg/kg and 5 mg/kg, respectively) and death was confirmed by absence of all the reflexes and by exsanguination. The heart was then isolated and either snap frozen in liquid nitrogen and kept at −80°C until use or fixed with 4% paraformaldehyde (pH 7.0) and then embedded in paraffin for immunofluorescence staining experiments. Experimental work with animals was approved by the Ethical Committee of the Veterinary Office of Fribourg Switzerland (2020–01-FR) and performed in compliance with guidelines on animal experimentation at our institution and in accordance with the updated ARRIVE guidelines (*Percie du Sert et al., 2020*). Experiments with human heart biopsies were approved by the Ethics Committee of Northwestern and Central Switzerland (Project-ID 2021–01445). Rat heart tissues were obtained from a previous study (*Borrego et al., 2022*) approved by the Ethical Committee of the Veterinary Office of Fribourg Switzerland (2019–21-FR).

## Langendorff-perfused mouse heart preparation

Ex vivo functional assessments were performed by using an isolated heart system (Langendorff apparatus – EMKA technologies, France). Briefly, after euthanasia as above described, the hearts of wt and $Arg2^{-/-}$ male and female mice (20–22 months old) were excised, cannulated, and retrogradely perfused on the Langendorff system. 150 µl of heparin (2000 units/ml) was injected 15 min before the heart removal to prevent blood coagulation. Hearts trimmed of extracardiac tissue were weighed immediately after surgical removal, prior to cannulation, and heart weight to body weight ratio (HW/BW) was calculated as hypertrophy index. The aorta was cannulated on a shortened and blunted 21-gauge needle and perfusion initiated at a constant pressure of 80 mmHg. The hearts were perfused with a modified Krebs–Henseleit solution composed of (mmol/L): 118 NaCl, 4.7 KCl, 1.2 $MgSO_4$, 1.2 $KH_2PO_4$, 25 $NaHCO_3$, 11 Glucose, 2 $CaCl_2$. The fluid was bubbled with a mix of 95% $O_2$ and 5% $CO_2$ at 37°C to give a pH of 7.4 and was filtered through an in-line 0.45 µm Sterivex-HV filter before delivery to the heart. A fluid-filled balloon was introduced into the left ventricle through an incision in the left atrial appendage. The ventricular balloon was connected via fluid-filled tubing to a pressure transducer for continuous assessment of ventricular function. The balloon was inflated to yield a left ventricular end-diastolic pressure between 5–15 mmHg. Hearts were immersed in warmed perfusate in a jacketed bath maintained at 37 °C, and perfusate delivered to the coronary circulation was maintained at the same temperature. After 20 min stabilization, hearts were subjected to 15 min (males), 20 min, and 30 min (females) of no-flow ischemia followed by 30–45 min of reperfusion. Recovery of contraction and relaxation rates ($dt/dp_{min}$) as well as the recovery of the left ventricle developed pressure (LV-DP) and left ventricular end diastolic pressure (LV-EDP) were measured to investigate tolerance to ischemia/reperfusion (I/R)-injury. In addition to functional recovery, triphenyl tetrazolium chloride (TTC) staining was also performed to investigate a potential reduction in infarct size upon I/R injury in $Arg2^{-/-}$ animals. TTC is a colorless water-soluble dye that is reduced by the mitochondrial enzyme succinate dehydrogenase of living cells into a water-insoluble, light-sensitive compound (formazan) that turns healthy/normal tissue deep red. In contrast, damaged/dead tissue remains white, showing the absence of living cells and thereby indicating the infarct region (*Bederson et al., 1986*). After completing the I/R protocol, the heart was carefully removed from the experimental set-up. After a freeze-thaw cycle (–20 °C for 1 hr), the heart was cross-sectioned from the apex to the atrioventricular groove into four 2.5-mm-thick slices. The slices were first incubated in TTC (20 min at 37 °C) and then fixed in 10% formalin for 30 min. A cover glass was then placed over the tissues and images were acquired (*Redfors et al., 2012*). ImageJ software was used to analyze the images and assess the infarcted area.

## Real-time quantitative RT-PCR

mRNA expression level of genes from mouse and human origin was measured by two-step quantitative Real Time-PCR as described previously (*Ming et al., 2012*). Ribosomal Protein S12 (*Rps12*) and glyceraldehyde-3-phosphate dehydrogenase (*Gapdh*) were used as reference genes. Total RNA from mouse tissue, splenic macrophages, and human cardiac fibroblasts was extracted with Trizol Reagent (TR-118, Molecular Research Center) following the manufacturer's protocol. Real-time PCR reaction was performed with the GOTaq qPCR Master Mix (A6001, Promega) and CFX96 Real-Time PCR Detection System (Bio-Rad). The mRNA expression level of all genes was normalized to the reference gene *Rps12* or *Gapdh*. All the qRT-PCR primer sequences are shown in *Supplementary file 2*.

## Confocal immunofluorescence microscope

Immunofluorescence staining was performed with heart tissues. Briefly, mouse hearts or human cardiac biopsies were fixed with 4% paraformaldehyde (pH 7.0) and embedded in paraffin. After deparaffinization in xylene (2 times of 10 min), the sections were treated in ethanol (twice in 100% ethanol for 3 min and twice in 95% ethanol and once in 80%, 75%, 50% ethanol for 1 min sequentially) followed by antigen retrieval (EDTA buffer, pH 8.0) for ARG2, TNNT, CD31, p16, MAC-2, IL-1β, Vimentin (VIM), CD-68, PDGF-Rα and α-smooth muscle actin (αSMA) in a pressure cooker. For co-immunofluorescence staining of ARG2/TNNT, ARG2/CD31, ARG2/VIM, Arg2/CD-68, ARG2/MAC-2, ARG2/PDGF-Rα, ARG2/αSMA, IL-1β/MAC-2, F4-80/CCR2, F4-80/LYVE1 (*Figure 2—figure supplement 1* shows the validation of CCR2 and LYVE1 antibodies) and CD31/VIM, primary antibodies of different species were used. Transverse sections (5 μm) were blocked with mouse Ig blocking reagent (M.O.M, Vector laboratories) for 2 hr and then with PBS containing 1% BSA and 10% goat serum for 1 hr. For CD31 and PDGF-Rα (both polyclonal Goat IgGs), a blocking reagent containing 10% BSA was used. The sections were then incubated overnight at 4°C in a dark/humidified chamber with target primary antibodies and subsequently incubated for 2 hr with the following secondary antibodies: Alexa Fluor 488–conjugated goat anti-rabbit IgG (H + L) and Alexa Fluor 568-conjugated goat anti-mouse IgG (H + L); Alexa Fluor 488–conjugated goat anti-mouse IgG (H + L) and Alexa Fluor 594-conjugated goat anti-rabbit IgG (H + L); Alexa Fluor 488–conjugated donkey anti-goat IgG (H + L) and Alexa Fluor 594-conjugated goat anti-rabbit IgG (H + L). All sections were finally counterstained with 300 nmol/l DAPI for 5 min. Immunofluorescence signals were visualized under Leica TCS SP5 confocal laser microscope. The antibodies used are shown in *Supplementary file 3*.

## Isolation and cultivation of ventricular cardiomyocytes

Ventricular myocytes were isolated from 5-month-old female mice according to an established protocol (*Louch et al., 2011*). Briefly, the hearts were excised, cannulated, and retrogradely perfused on a Langendorff system as described above. 150 μl of heparin (2000 units/ml) was injected 15 min before the heart removal to prevent blood coagulation. To isolate the cardiomyocytes, hearts were perfused at 37°C for 20 min with a $Ca^{2+}$-free solution composed of (in mmol/ l): 130 NaCl, 5 KCl, 0.5 $NaH_2PO_4$, 10 HEPES, 10 Glucose, 10 2.3 butanedione monoxime (BDM), 10 Taurine, 1 $MgCl_2$ (pH 7.4, adjusted with NaOH). Cells were enzymatically dissociated using a cocktail of collagenase type II (210 U/ml, Worthington), collagenase type IV (260 U/ml, Worthington), and protease type XIV (0.21 U/ml; Sigma-Aldrich). After isolation, cardiomyocytes were separated by gravity settling and $Ca^{2+}$ was restored to a final concentration of 1 mmol/l. The cardiomyocytes was finally plated onto laminin-coated coverslips (5 μg/ml, Thermo Scientific) and cultured in the M-199 medium supplemented with bovine serum albumin (BSA; 0.1% - A1470, Sigma-Aldrich), BDM (10 mmol/l), insulin, transferrin, selenium (ITS; I3146 - Sigma-Aldrich), CD lipid (11905–031 - Sigma-Aldrich) and penicillin / streptomycin mix (1 %).

## Cell culture, adenoviral transduction, and Nos2[-/-] and Arg2[-/-] macrophages

Human cardiac fibroblasts (HCF) from adult human heart tissue were purchased from Innoprot (P10452) and cultured in the proprietary fibroblast culture medium (P60108-2, Innoprot) composed as follows: 500 ml of fibroblast basal medium, 25 ml of FBS, 5 ml of fibroblast growth supplement-2, and 5 ml of penicillin/streptomycin in the Poly-L-Lysine (PLL) coated flasks and dishes. The cells were maintained at 37°C in a humidified incubator (5% $CO_2$ and 95% atmosphere air). To overexpress *Arg2*, the cells were seeded on 6 cm dishes for 24 hr and transduced with rAd-CMV-Con/ *Arg2* for 48 hr as described previously (*Ming et al., 2012*).

Human umbilical vein endothelial cells (HUVEC) were cultured in RPMI-1640 (with 25 mmol/l HEPES and stable glutamine; Thermo Fisher Scientific) containing 5% fetal calf serum (FCS; Gibco) and 1% streptomycin and penicillin in gelatin (1%) coated dishes.

Raw 264.7 cells (mouse macrophage cell line) were cultured in Dulbecco modified Eagle medium/ F12 (DMEM/F12; Thermo Fisher Scientific) containing 10% fetal bovine serum (FBS; Gibco) and 1% streptomycin and penicillin. To silence *Arg2*, the cells were seeded on 6 cm dishes for 24 hr and transduced first with the rAd at titers of 100 Multiplicity of Infection (MOI) and cultured in the complete medium for 2 days and then in serum-free medium for another 24 hr before experiments. The rAd expressing shRNA targeting *Arg2* driven by the U6 promoter (rAd/U6-arg-ii-shRNA) and control rAd

expressing shRNA targeting *Lacz* (rAd/U6-*lacz*-shRNA) were generated as described previously (***Ming et al., 2009***).

Wt and *Nos2* gene-deficient bone-marrow-derived macrophage cell lines (MØ$^{wt}$ and MØ$^{Nos2-/-}$) were purchased from Kerafast (ENH166-FP and ENH176-FP) and cultured in Dulbecco modified Eagle medium (DMEM; Thermo Fisher Scientific) containing 10% fetal bovine serum (FBS; Gibco) and 1% streptomycin and penicillin. The cells were treated with LPS (100 ng/ml) for 24 hr before harvest. Gene deficiency of *Nos2* was verified by immunoblotting analysis.

Knockout of the *Arg2* gene in human THP-1 cells was generated by CRISPR-UTM-mediated genome engineering with gRNA targeting exon 1 (Guangzhou Ubigene Biosciences Co., Ltd.). The control THP1 cell line was generated with scramble gRNA and Cas9. Gene deficiency of *Arg2* was verified by immunoblotting analysis.

All cell lines used in this study (RAW 264.7, THP-1 wt and *Arg2* knockout, and BMDMs wt and *Nos2* knockout) were routinely tested and confirmed to be negative for mycoplasma contamination throughout the experimental period.

Short tandem repeat (STR) profiling was performed through Microsynth for all cell lines. STR profiles for THP-1 and RAW 264.7 demonstrated excellent concordance (≥98.6%) with reference cell lines in the Cellosaurus database, confirming their identity and excluding contamination. The full STR datasets are provided as supplementary data (***Supplementary file 4***, ***Supplementary file 5***, ***Supplementary file 6*** for RAW 264.7, THP-1 wt, and THP-1 *Arg2* knockout, respectively).

For the BMDMs (wt and *Nos2* knockout), STR analysis confirmed murine origin, absence of human DNA contamination, and a 100% match between wt and knockout cells, consistent with a shared genetic background. However, no external STR reference profile is currently available for these cells, as confirmed by both the supplier (Kerafast) and the Cellosaurus database. Therefore, direct validation against a reference cannot be carried out at this time. STR profiles are provided as supplementary data (***Supplementary file 7*** and ***Supplementary file 8*** for BMDMs wt and BMDMs *Nos2* knockout, respectively).

## TUNEL Staining

Mouse hearts were isolated and fixed with 4% paraformaldehyde (pH 7.0), embedded in paraffin and sliced at 5 μm of thickness. After deparaffinization (xylene, 2 times for 10 min) and rehydration (sequential washing in 100%, 95%, 80%, 75%, and 50% ethanol), TUNEL staining was performed using a commercial kit (in situ Cell Death Detection Kit, TMR red – Roche, Switzerland). Briefly, 50 μl of TUNEL mixture, containing TdT and dUTP in reaction buffer, were added onto the sections and incubated in a humidified chamber for 60 min at 37°C in darkness. Positive controls were generated by incubating samples at room temperature (RT) for 10 min with DNase in PBS (1 mg/ml) to induce strand breaks. Negative controls were generated by incubation with TUNEL mixture lacking TdT. DAPI counterstaining was finally performed (300 nmol/l for 5 min).

TUNEL staining was also performed on isolated ventricular cardiomyocytes. After tissue digestion, the cardiomyocytes were plated onto murine laminin-coated coverslips to promote cell adhesion. One hour after plating, several PBS washing steps were performed to remove damaged and partially adherent cells, ensuring that only well-shaped, viable, and strongly adherent cells remained as bioassay cells. These 'healthy' cells were then selected for the experiments. After washing, cardiomyocytes were incubated for 24 hr with either splenic macrophage conditioned media (CM; see below) or with RAW cells CM, and the following protocol was applied: cells are fixed using 4% paraformaldehyde for 1 hr at RT, and then permeabilized via Triton X (0.1 %). As for tissues, 50 μl of TUNEL mixture, containing TdT and dUTP in reaction buffer, was added onto each slide and then incubated at 37°C for 60 min in darkness. DAPI counterstaining was then performed.

Upon plating onto murine laminin-coated coverslips, the cardiomyocytes were first incubated for 24 hr with either splenic macrophage conditioned media (CM; see below) or with RAW cells CM, and the following protocol was applied: cells are fixed using 4% paraformaldehyde for 1 hr at RT, and then permeabilized via Triton X (0.1 %). As for tissues, 50 μl of TUNEL mixture, containing TdT and dUTP in reaction buffer, were added onto each slide and then incubated at 37°C for 60 min in darkness. DAPI counterstaining was then performed.

For experiments aiming to define cell borders, TUNEL protocol was followed by a staining with Wheat Germ Agglutinin (WGA)-Alexa Fluor 488 (10 μg/ml for 30 min). For heart tissue, five

representative images of each section are taken using a Leica TCS SP5 confocal system, and WGA staining was performed to differentiate cardiomyocytes and non-cardiomyocytes according to cell size and presence or absence of striations (characteristics of cardiomyocytes). Data was reported as number of TUNEL-positive nuclei over the total number of nuclei in the images.

## Masson's trichrome staining

To analyze cardiac fibrosis, heart sections (5 µm) were subjected to Masson's trichrome (ab150686, Abcam) staining according to the manufacturer's instructions (*de Guzman et al., 2023*).

## Hydroxyproline colorimetric assay

Collagen content was analyzed by determination of hydroxyproline levels in left ventricles or in cell homogenates using the Hydroxyproline Assay kit (MAK008, Sigma) according to the manufacturer's instructions.

## Isolation of splenic macrophages

Splenic cells were isolated as previously described (*Wang et al., 2013*). Briefly, young and old wt and $Arg2^{-/-}$ female mice were euthanized as described and spleens were dissected from abdominal cavity and filtered through a 70 µm nylon strainer. Red blood cell lysis buffer was used to remove red blood cells. The cell suspension obtained was plated onto gelatin-coated dishes at a density of $2 \times 10^6$ cells/ml. After 3 days, cells were washed to ensure macrophage enrichment. Following washing, the macrophages were cultured for a further 4 days in RPMI culture media with 20% of L929 cell line conditioned medium (source of the macrophage growth factor M-CSF) to allow cell maturation. Medium was changed every two days.

## Crosstalk between macrophages and heart cells

For crosstalk experiments of splenic macrophages from wt and $Arg2^{-/-}$ mice with cardiac cells, conditioned medium from the isolated splenic macrophages was collected and transferred to cardiomyocytes, fibroblasts, and endothelial cells in 1:1 dilution with the specific cell culture medium for the indicated time. To study the effects of IL-1β in the conditioned medium of splenic macrophages (CM-SM), cardiac cells, fibroblasts, or endothelial cells were pre-treated with IL-1 receptor antagonist IL-1RA (50 ng/ml) for 2 hr followed by the specific experimental protocols for analysis of cardiac apoptosis, fibroblast activation, and EndMT transition, respectively.

For the experiments with mouse macrophage cell line Raw 264.7, the cells were seeded on six-well plates with a density of $2 \times 10^5$ cells per well. Upon transduction with rAd expressing $Arg2^{shRNA}$ (see above) and overnight serum starvation, the mouse macrophages were polarized toward a pro-inflammatory secretion profile by stimulation with lipopolysaccharide (LPS; 100 ng/ml) for 24 hr. The conditioned medium (CM) from non-treated control and LPS-activated RAW cells (referred to as CM-RAW) was then filtered and transferred onto cardiomyocytes for 24 hr to investigate cell apoptosis. To study the effects of IL-1β in CM-RAW on cardiac cells apoptosis, myocytes were pre-treated with IL-1 receptor antagonist IL-1RA (280-RA, R&D systems) for 2 hr.

## Hypoxia experiments with murine cardiac fibroblasts and cardiomyocytes

Cardiac cells were isolated from wt mice according to an established protocol (*Louch et al., 2011*). Briefly, after euthanasia of mice, hearts were excised, cannulated, and retrogradely perfused on a Langendorff system. Cells were enzymatically dissociated using a cocktail of collagenase type II, collagenase type IV, and protease type XIV. After isolation, cardiomyocytes were separated from the pull of cardiac cells by gravity settling and plated onto laminin-coated coverslips. After 2 hr, the cardiomyocytes were washed to remove non-adhering cells and incubated for 24 hr at 1% oxygen level in a Coy In Vitro Hypoxic Cabinet System (The Coy Laboratory Products, Grass Lake, MI USA). The rest of cardiac cells were plated on 12-well plates and cultured in DMEM (10% FBS) for 3 days, allowing cardiac fibroblasts to adhere. The fibroblasts were passaged once and subjected to hypoxia as for cardiomyocytes.

## Hypoxia experiments with human cardiac fibroblasts and mitochondrial superoxide detection (MitoSOX staining)

Human cardiac fibroblasts (HCF) from adult human heart tissue were incubated for 48 hr at 1% oxygen level in a Coy In Vitro Hypoxic Cabinet System (The Coy Laboratory Products, Grass Lake, MI USA). To silence *Arg2*, HCFs were seeded on 6 cm dishes for 24 hr and transduced first with the rAd at titers of 100 Multiplicity of Infection (MOI) and cultured in the complete medium for 2 days and then in serum-free medium for another 24 hr before experiments. The rAd expressing shRNA targeting *Arg2* driven by the U6 promoter (rAd/U6- *Arg2*-shRNA) and control rAd expressing shRNA targeting *Lacz* (rAd/U6-*Lacz*-shRNA) were generated as described previously (*Ming et al., 2009*). Mitochondrial superoxide generation was monitored by MitoSOX. Briefly, the cells were incubated with MitoSOX at the concentration of 1 μmol/l for 30 min. After washing, the cells were then fixed with 3.7% paraformaldehyde followed by counterstaining with DAPI and then subjected to imaging through 40×objectives with Leica TCS SP5 confocal laser microscope. To ensure the mitochondria as a source of the ROS, cells were treated with Mito TEMPO (10 μmol/l, 1 hr) followed by MitoSOX staining.

## ELISA for IL-1β detection in conditioned media

IL-1β concentrations in the conditioned medium from murine splenic cells were measured by ELISA kits (mouse IL-1β, 432604, BioLegend, San Diego, USA) according to the manufacturer's instructions. IL-1β concentrations in the conditioned medium from THP1 human cells were measured by ELISA kits (human IL-1β, 437004, BioLegend, San Diego, USA) according to the manufacturer's instructions.

## Immunoblotting

Heart tissue or cell lysate preparation, SDS-PAGE and immunoblotting, antibody incubation, and signal detection were performed as described previously (*Ming et al., 2012*). To prepare heart homogenates, frozen cardiac tissues were crushed into the fine powder using a mortar and pestle in liquid nitrogen on ice. A portion of the fine powder was then homogenized (XENOX-Motorhandstück MHX homogenizer) on ice in 150 μl of ice-cold lysis buffer with the following composition: 10 mmol/l Tris-HCl (pH 7.4), 0.4% Triton X-100, 10 μg/ml leupeptin, and 0.1 mmol/l phenylmethylsulfonyl fluoride (PMSF), protease inhibitor cocktail (B14002, Bio-Tool) and phosphatase inhibitor cocktail (B15002; Bio-Tool). Homogenates were centrifuged (Sorvall Legend Micro 17 R) at $13,000 \times g$ for 15 min at 4°C. Protein concentrations of supernatants were determined by Lowry method (500–0116, Bio-Rad). An equal amount of protein from each sample was heated at 75°C for 15 min in loading buffer and separated by SDS-PAGE electrophoresis. Proteins in the SDS-PAGE gel were then transferred to PVDF membranes which are blocked with PBS-Tween-20 supplemented with 5% skimmed milk. The membranes were then incubated with the corresponding primary antibody overnight at 4°C with gentle agitation. After washing with blocking buffer, the membranes were then incubated with the corresponding anti-mouse or anti-rabbit secondary antibody. Signals were visualized using the Odyssey Infrared Imaging System (LI-COR Biosciences), or the FUSION FX Imaging system (Witec AG) for chemiluminescence, and quantified by Image Studio Lite (5.2, LI-COR Biosciences). The antibodies used in this study are listed in the *Supplementary file 3*.

## Statistical analysis

Data was presented as mean ± SD. Data distribution was determined by Kolmogorov-Smirnov test and statistical analysis was performed by using GraphPad Prism software. Student's T test or analysis of variance (ANOVA) with post hoc test was used to test significance. Differences in mean values were considered statistically significant at a two-tailed p-value≤0.05.

# Additional information

### Funding

| Funder | Grant reference number | Author |
| --- | --- | --- |
| Swiss National Science Foundation | 31003A_179261/1 | Zhihong Yang |

| Funder | Grant reference number | Author |
|---|---|---|
| Swiss Heart Foundation | FF19033 | Xiu-Fen Ming |

The funders had no role in study design, data collection and interpretation, or the decision to submit the work for publication.

## Author contributions

Duilio M Potenza, Conceptualization, Data curation, Formal analysis, Validation, Investigation, Methodology, Writing – original draft, Writing – review and editing; Xin Cheng, Guillaume Ajalbert, Data curation, Formal analysis, Validation, Investigation, Methodology, Writing – review and editing; Andrea Brenna, Writing – review and editing; Marie-Noelle Giraud, Investigation, Writing – review and editing; Aurelien Frobert, Investigation, Methodology, Writing – review and editing; Stephane Cook, Kirsten D Mertz, Resources, Writing – review and editing; Zhihong Yang, Conceptualization, Resources, Data curation, Supervision, Funding acquisition, Validation, Investigation, Writing – original draft, Project administration, Writing – review and editing; Xiu-Fen Ming, Conceptualization, Resources, Data curation, Formal analysis, Supervision, Funding acquisition, Investigation, Writing – original draft, Project administration, Writing – review and editing

## Author ORCIDs

Duilio M Potenza ⓘ https://orcid.org/0000-0003-2442-9154
Andrea Brenna ⓘ https://orcid.org/0000-0002-8542-9855
Zhihong Yang ⓘ https://orcid.org/0000-0002-4133-5099
Xiu-Fen Ming ⓘ https://orcid.org/0000-0002-5848-4496

## Ethics

Experimental work with animals was approved by the Ethical Committee of the Veterinary Office of Fribourg Switzerland (2020-01-FR, and 2019-21-FR) and performed in compliance with guidelines on animal experimentation at our institution. The study was conducted according to the principles expressed in the Declaration of Helsinki.

Reviewer #1 (Public review): https://doi.org/10.7554/eLife.94794.3.sa1
Reviewer #2 (Public review): https://doi.org/10.7554/eLife.94794.3.sa2
Author response https://doi.org/10.7554/eLife.94794.3.sa3

# Additional files

## Supplementary files

Supplementary file 1. Baseline comparison between *wt* and *Arg2*-/- mice under Langendorff recordings.

Supplementary file 2. List of RT-PCR primer sequences.

Supplementary file 3. List of antibodies used in the study with specific dilutions for immunoblotting and immunofluorescence staining.

Supplementary file 4. Short tandem repeat (STR) profiling of RAW 264.7, which shows 98.6% concordance with reference cell lines in the Cellosaurus database, confirming the cell line identity.

Supplementary file 5. Short tandem repeat (STR) profiling of THP1 wt, which shows 100% concordance with reference cell lines in the Cellosaurus database, confirming the cell line identity.

Supplementary file 6. Short tandem repeat (STR) profiling of THP1 *Arg2* knockout, which shows 100% concordance with reference cell lines in the Cellosaurus database, confirming the cell line identity.

Supplementary file 7. Short tandem repeat (STR) profiling of wt bone-marrow-derived macrophage cell line.

Supplementary file 8. Short tandem repeat (STR) profiling of *Nos2* knockout bone-marrow-derived macrophage cell line.

MDAR checklist

## Data availability

All datasets generated and analyzed during this study (including western blot images) have been deposited in the Dryad Digital Repository and are publicly available at https://doi.org/10.5061/dryad. hx3ffbgrt. Further details regarding the dataset content, structure, and data organization are fully described within the Dryad entry.

The following dataset was generated:

| Author(s) | Year | Dataset title | Dataset URL | Database and Identifier |
|---|---|---|---|---|
| Potenza DM, Cheng X, Ajalbert G, Brenna A, Giraud M, Frobert A, Cook S, Mertz K, Yang Z, Ming X | 2025 | Cell-autonomous and non-cell-autonomous effects of Arginase 2 on cardiac aging | https://doi.org/10.5061/dryad.hx3ffbgrt | Dryad Digital Repository, 10.5061/dryad.hx3ffbgrt |

The following previously published dataset was used:

| Author(s) | Year | Dataset title | Dataset URL | Database and Identifier |
|---|---|---|---|---|
| Wolff C, Gutierrez-Monreal M, Meng L, Zhang X, Douma L, Costello H, Douglas C, Ebrahimi E, Alava B, Morris A, Endale M, Crislip G, Cheng K, Schroder E, Delisle B, Bryant A, Gumz M, Huo Z, Liu A, Esser K | 2023 | Defining the age-dependent and tissue-specific circadian transcriptome in male mice | https://www.ncbi.nlm.nih.gov/geo/query/acc.cgi?acc=GSE201207 | NCBI Gene Expression Omnibus, GSE201207 |

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
