## [Editor Report · eLife Assessment]

This study provides **fundamental** information on how Arg-II participates in cardiac aging. The phenotypic data provide **convincing** evidence of non-cell-autonomous contributions to aging-related pathologies. Overall, the study highlights the importance of intercellular signaling in maintaining cardiac health during aging.

---

## [Referee Report · Reviewer #1 (Public review)]

Summary:

The manuscript by Duilio M. Potenza et al. explores the role of Arginase II in cardiac aging, majorly using whole-body arg-ii knock-out mice. In this work, the authors have found that Arg-II exerts non-cell-autonomous effects on aging cardiomyocytes, fibroblasts, and endothelial cells mediated by IL-1b from aging macrophages. The authors have used arg II KO mice and an in vitro culture system to study the role of Arg II. Authors have also reported the cell-autonomous effect of Arg-II through mitochondrial ROS in fibroblasts that contribute to cardiac aging. These findings are sufficiently novel in cardiac aging and provide interesting insights. While the phenotypic data seem strong, the mechanistic details are unclear. How Arg II regulates the IL-1b and modulates cardiac aging is still being determined.

Strengths:

This study provides interesting information on the role of Arg II in cardiac aging.

The phenotypic data in the Arg II KO mice is convincing, and the authors have assessed most of the aging-related changes.

The data is supported by an in vitro cell culture system.

Weaknesses:

The manuscript needs more mechanistic details on how Arg II regulates IL-1b and modulates cardiac aging.

---

## [Referee Report · Reviewer #2 (Public review)]

This study investigates the role of arginase-II (Arg-II) in cardiac aging. The authors challenge previous assumptions by demonstrating that Arg-II is not expressed in aged cardiomyocytes, but is upregulated in non-myocyte cells, specifically macrophages, fibroblasts, and endothelial cells. Using Arg-II knockout mice, they show protection against age-associated cardiac inflammation, fibrosis, apoptosis, endothelial-to-mesenchymal transition (EndMT), and ischemic injury. Mechanistically, Arg-II promotes IL-1β release from macrophages and increases mitochondrial ROS in fibroblasts, contributing to cardiac aging through both cell-autonomous and non-cell-autonomous mechanisms.

The study is well-structured and combines genetic models, molecular assays, and histological analyses to support its conclusions. Including both human and mouse samples strengthens the translational relevance of the findings. The authors have addressed most of the reviewers' comments and have made efforts to improve the manuscript by adding experimental data, explanations, and further discussion.

The data convincingly support their conclusions. This work provides valuable insights into the mechanisms of cardiac aging, aligns with growing evidence of non-cell-autonomous contributions to aging-related pathologies, and highlights the importance of intercellular signaling in maintaining cardiac health during aging.

Although the use of cell-specific knockout mouse models would enhance the depth and translational potential of the findings, it is understandable that such an approach would be beyond the scope of a single study. This work lays the groundwork for future investigations into conditional Arg-II knockouts in specific cell types to elucidate the cell-specific roles of Arg-II in cardiac aging.

Overall, this is a solid and impactful study with strong experimental support

---

## [Author Response]

The following is the authors’ response to the original reviews

**Public Reviews:**

**Reviewer #1 (Public Review):**
Summary:The manuscript by Duilio M. Potenza et al. explores the role of Arginase II in cardiac aging, majorly using whole-body arg-ii knock-out mice. In this work, the authors have found that Arg-II exerts non-cell-autonomous effects on aging cardiomyocytes, fibroblasts, and endothelial cells mediated by IL-1b from aging macrophages. The authors have used arg II KO mice and an in vitro culture system to study the role of Arg II. The authors have also reported the cell-autonomous effect of Arg-II through mitochondrial ROS in fibroblasts that contribute to cardiac aging. These findings are sufficiently novel in cardiac aging and provide interesting insights. While the phenotypic data seems strong, the mechanistic details are unclear. How Arg II regulates the IL-1b and modulates cardiac aging is still being determined. The authors still need to determine whether Arg II in fibroblasts and endothelial contributes to cardiac fibrosis and cell death. This study also lacks a comprehensive understanding of the pathways modulated by Arg II to regulate cardiac aging.

We sincerely appreciate the valuable feedback provided by the reviewer. It's gratifying to hear that our work provided novel information on the role of arginase-II in cardiac aging which is a complex process involving various cell types and mechanisms. We have devoted considerable effort by performing new experiments to address the reviewer's comments and to delineate more detailed mechanisms of Arg-II in cardiac aging. Please, see below our specific answers to each point of the reviewers.

Strengths:This study provides interesting information on the role of Arg II in cardiac aging.The phenotypic data in the arg II KO mice is convincing, and the authors have assessed most of the aging-related changes.The data is supported by an in vitro cell culture system.

We appreciate this reviewer’s positive assessment on the strength of our study.

Weaknesses:The manuscript needs more mechanistic details on how Arg II regulates IL-1b and modulates cardiac aging.

We made great effort and have performed new experiments in human monocyte cell line (THP1) in which iNOS is not expressed and not inducible by LPS and *arg-ii* gene was knocked out by CRISPR technology. Moreover, murine bone-marrow derived macrophages in which *inos* gene was ablated, is also use for this purpose. We found that in the human THP1 monocytes in which Arg-II but not iNOS is induced by LPS (100 ng/mL for 24 hours) (Suppl. Fig. 6A), mRNA and protein levels of IL-1b precursor are markedly reduced in *arg-ii* knockout THP1*arg-ii-/-* as compared to the THP1*wt* cells (Suppl. Fig. 6B and 6C), further confirming that Arg-II promotes IL-1b production as also shown in RAW264.7 macrophages (Suppl. Fig. 5A and 5C). Moreover, in the mouse bone-marrow-derived macrophages, LPS-induced IL-1b production is inhibited by *inos* deficiency (BMDM*inos-/-* vs BMDM*wt*) (Suppl. Fig. 6D and 6E), while Arg-II levels are slightly enhanced in the BMDM*inos-/-* cells (Suppl. Fig. 6D and 6F). All together, these results suggest that iNOS slightly reduces Arg-II expression. Arg-II and iNOS can be upregulated by LPS independently. Both Arg-II and iNOS are required for IL-1b production upon LPS stimulation as illustrated in Suppl. Fig. 6G. For detailed results and discussion, please see answers to the comments point 2 or point 6 raised by this reviewer.

The authors used whole-body KO mice, and the role of macrophages in cardiac aging is not studied in this model. A macrophage-specific arg II Ko would be a better model.

We fully agree with this comment of the reviewer. Unfortunately, this macrophage specific *arg-ii* knockout animal model is not available, yet. Future research shall develop the macrophage-specific *arg-ii-/-* mouse model to confirm this conclusion with aging animals. Since Arg-II is also expressed in fibroblasts and endothelial cells and exerts cell-autonomous and paracrine functions, aging mouse models with conditional *arg-ii* knockout in the specific cell types would be the next step to elucidate cell-specific function of Arg-II in cardiac aging. We have pointed out this aspect for future research on page 19, lines 2 to 6.

Experiments need to validate the deficiency of Arg II in cardiomyocytes.

As pointed out by this reviewer in the comment point 10, Arg-II was previously reported to be expressed in isolated cardiomyocytes from in rats (PMID: 16537391). Unfortunately, negative controls. i.e., *arg-ii-/-* samples were not included in the study to avoid any possible background signals. We made great effort to investigate whether Arg-II is present in the cardiomyocytes from different species including mice, rats and humans and have included old *arg-ii-/-* mouse samples as a negative control. This allows to validate the antibody specificity and background noises beyond any reasonable doubt. The new experiments in Suppl. Fig. 4 confirms the specificity of the antibody against Arg-II in old mouse kidney which is known to express Arg-II in the S3 proximal tubular cells (Huang J, et al. 2021). To exclude the possible species-specific different expression of Arg-II in the cardiomyocytes, aged mouse and rat heart tissues were used for cellular localization of Arg-II by confocal immunofluorescence staining. As shown in Suppl. Fig. 4B and 4C, both species show Arg-II expression only in non-cardiomyocytes (cells between striated cardiomyocytes) (red arrows) but not in striated cardiomyocytes. Even in the rat myocardial infarction tissues, Arg-II was not found in cardiomyocytes but in endocardium cells (Suppl. Fig. 4B). In isolated cardiomyocytes exposed to hypoxia, a well know strong stimulus for Arg-II protein levels, no Arg-II signals could be detected, while in fibroblasts from the same animals, an elevated Arg-II levels under hypoxia is demonstrated (Fig. 5B). Furthermore, even RT-qPCR could not detect *arg-ii* mRNA in cardiomyocytes but in non-cardiomyocytes (Fig. 5C). All together, these results demonstrate that Arg-II are not expressed or at negligible levels in cardiomyocytes but expressed in non-cardiomyocytes. This new experiments with rat heart are included in the method section on page 20, the 1st paragraph. The results are described on page 7, the 1st paragraph, and discussed on page 12, the 2nd paragraph. Legend to Suppl. Fig. 4 is included in the file “Suppl. figure legend_R”.

The authors have never investigated the possibility of NO involvement in this mice model.

As above mentioned, we made great effort and have performed new experiments in human monocyte cell line (THP1) in which iNOS is not expressed and not inducible by LPS and *arg-ii* gene was knocked out by CRISPR technology. Moreover, murine bone-marrow derived macrophages in which *inos* gene was ablated, is also use for this purpose. The results show that Arg-II and iNOS can be upregulated by LPS independent of each other and iNOS slightly reduces Arg-II expression. However, both Arg-II and iNOS are required for IL-1b production upon LPS stimulation. For detailed results and discussion, please see answers to the comments point 2 or point 6 raised by this reviewer.

A co-culture system would be appropriate to understand the non-cell-autonomous functions of macrophages.

We appreciate the suggestion by this reviewer regarding the co-culture system to test the non-cell autonomous role of Arg-II. We think that our current model, which involves treating cells with conditioned media, is a well-established and effective method for demonstrating the non-cell autonomous role of Arg-II. This approach allows us to observe the effects of Arg-II on surrounding cells through the factors present in the conditioned media released from macrophages. The co-culture system could be considered, if the released factor in the conditioned medium is not stable. This is however not the case. Therefore, we are confident that our experimental model with conditioned medium is sufficiently enough to demonstrate a paracrine effect of cell-cell interaction (please also see answers to the comment point 16).

The Myocardial infarction data shown in the mice model may not be directly linked to cardiac aging.

As we have introduced and discussed in the manuscript, aging is a predominant risk factor for cardiovascular disease (CVD). Studies in experimental animal models and in humans provide evidence demonstrating that aging heart is more vulnerable to stressors such as ischemia/reperfusion injury and myocardial infarction as compared to the heart of young individuals. Even in the heart of apparently healthy individuals of old age, chronic inflammation, cardiomyocyte senescence, cell apoptosis, interstitial/perivascular tissue fibrosis, endothelial dysfunction and endothelial-mesenchymal transition (EndMT), and cardiac dysfunction either with preserved or reduced ejection fraction rate are observed. Our study is aimed to investigate the role of Arg-II in cardiac aging phenotype and age-associated cardiac vulnerability to stressors. Therefore, cardiac functional changes and myocardial infarction in response to ischemia/reperfusion injury are suitable surrogate parameters for the purpose.

**Reviewer #2 (Public Review):**
Summary:The results from this study demonstrated a cell-specific role of mitochondrial enzyme arginase-II (Arg-II) in heart aging and revealed a non-cell-autonomous effect of Arg-II on cardiomyocytes, fibroblasts, and endothelial cells through the crosstalk with macrophages via inflammatory factors, such as by IL-1b, as well as a cell-autonomous effect of Arg-II through mtROS in fibroblasts contributing to cardiac aging phenotype. These findings highlight the significance of non-cardiomyocytes in the heart and bring new insights into the understanding of pathologies of cardiac aging. It also provides new evidence for the development of therapeutic strategies, such as targeting the ArgII activation in macrophages.

We're grateful for the reviewer's positive feedback, acknowledging the significant findings of our study on the role of arginase-II (Arg-II) in cardiac aging. We appreciate this reviewer’s insight into the therapeutic potential of targeting Arg-II activation in macrophages and are excited about the implications for future interventions in age-related cardiac pathologies. Thank you for recognizing the importance of our work in advancing our understanding of cardiac aging and potential therapeutic strategies.

Strengths:This study targets an important clinical challenge, and the results are interesting and innovative. The experimental design is rigorous, the results are solid, and the representation is clear. The conclusion is logical and justified.

We thank this reviewer for the positive comment.

Weaknesses:The discussion could be extended a little bit to improve the realm of the knowledge related to this study.

We appreciate this comment and have added and revised our discussion on this aspect accordingly at the end of the discussion section on page 19.

**Recommendations for the authors:**

**Reviewer #1 (Recommendations For The Authors):**
I have several critical concerns, specifically about the mechanism of how Arg-II plays a role in cardiac aging.My major concerns are:(1) The authors have shown non-cell-autonomous effects on aging cardiomyocytes, fibroblasts, and endothelial cells mediated by IL-1b from aging macrophages. A macrophage-specific Arg-II knock-out mouse model is a suitable and necessary control to establish claims.

We fully agree with this comment of the reviewer. Unfortunately, this macrophage specific *arg-ii* knockout animal model is not available, yet. Future research shall develop the macrophage-specific *arg-ii-/-* mouse model to confirm this conclusion with aging animals. Since Arg-II is also expressed in fibroblasts and endothelial cells and exerts cell-autonomous and paracrine functions, aging mouse models with conditional *arg-ii* knockout in the specific cell types would be the next step to elucidate cell-specific function of Arg-II in cardiac aging. We have pointed out this aspect for future research on page 19, lines 2 to 6.

(2) This study suggests that Arg-II exerts its effect through IL-1b in cardiac ageing. However, all experiments performed to demonstrate the link between ArgII and IL-1β are correlative at best. The underlying molecular mechanism, including transcription factors involved in the regulation of IL-1β by *arg-ii*, has not been demonstrated.

We sincerely appreciate this reviewer’s comment on the aspect! To make it clear, a causal role of Arg-II in promoting IL-1β production in macrophages is evidenced by the experimental results showing that old *arg-ii-/-* mouse heart has lower IL-1β levels than the age-matched *wt* mouse heart (Fig. 6A to 6D). We further showed that the cellular IL-1β protein levels and release are reduced in old *arg-ii-/-* mouse splenic macrophages as compared to the *wt* cells (Fig. 7A, 7C, and 7D). This result is further confirmed with the mouse macrophage cell line RAW264.7 (Suppl. Fig. 5A and suppl. Fig. 5C), in which we demonstrate that silencing *arg-ii* reduces IL-1β levels stimulated with LPS.

According to this reviewer’s comment (see comment point 6), we made further effort to investigate possible involvement of iNOS in Arg-II-regulated IL-1β production in macrophages stimulated with LPS. We performed new experiments in human monocyte cell line (THP1) in which iNOS is not expressed and not inducible by LPS and *arg-ii* gene was knocked out by CRISPR technology in the cells.

Moreover, murine bone-marrow derived macrophages in which *inos* gene was ablated, is also use for this purpose. We found that in the human THP1 monocytes in which Arg-II but not iNOS is induced by LPS (100 ng/mL for 24 hours) (Suppl. Fig. 6A), mRNA and protein levels of IL-1b are markedly reduced in *arg-ii* knockout THP1*arg-ii-/-* as compared to the THP1*wt* cells (Suppl. Fig. 6B and 6C), further confirming that Arg-II promotes IL-1b production as also shown in RAW264.7 macrophages (Suppl. Fig. 5A and 5C). The results suggest that Arg-II promotes IL-1b production independently of iNOS. Moreover, the role of iNOS in IL-1b production was also studied in the mouse bone-marrow-derived macrophages in which *inos* gene is ablated. The results demonstrate that LPS-induced IL-1b production is inhibited by *inos* deficiency (BMDM*inos-/-* vs BMDM*wt*) (Suppl. Fig. 6D and 6E), while Arg-II levels are slightly enhanced in the BMDM*inos-/-* cells (Suppl. Fig. 6D and 6F). Since arginase and iNOS share the same metabolic substrate L-arginine, *inos-/-* is expected to increase IL-1b production. This is however not the case. A strong inhibition of IL-1β production in *inos-/-* macrophages is observed. These results implicate that iNOS promotes IL-1β production independently of Arg-II and the inhibiting effect of IL-1β by *inos* deficiency is dominant and able to counteract Arg-II’s stimulating effect on IL-1β production. Hence, our results demonstrate that Arg-II promotes IL-1β production in macrophages independently of iNOS. All together, these results suggest that iNOS slightly reduces Arg-II expression. Arg-II and iNOS can be upregulated by LPS independently. Both Arg-II and iNOS are required for IL-1b production upon LPS stimulation (This concept is illustrated in the Suppl. Fig. 6G). The new results are described on page 8, the last paragraph and page 9, the 1st paragraph, presented in Suppl. Fig.6. The legend to Suppl. Fig. 6 is described in the file “Supplementary figure legend-R”. The related experimental methods are updated on page 23, the last two paragraphs and page 26 the last paragraph. The results are discussed o page 14, the last paragraph and page 15, the first two paragraphs.

(3) Figure 2: The authors have not validated the whole-body Arg-II knock-out mice for arg-ii ablation.

Thanks for pointing out this missing information! We have added the information regarding genotyping of the mice in the method section on page 20, first paragraph. Moreover, Fig. 5C also confirms the genotyping of the non-cardiomyocyte cells isolated from *wt* and *arg-ii-/-* animals.

(4) It is unclear why the authors have chosen to focus on IL-1β specifically, among other pro-inflammatory cytokines that were also downregulated in Arg-II-/- mice as demonstrated in Fig. 2A-D.

We appreciate the reviewer's question, which provides an opportunity to delve deeper into our findings. In our investigation, we observed that aging is accompanied by elevated levels of various proinflammatory markers. Intriguingly, our data revealed that *tnf-α* remained unaffected by the ablation of *arg-ii* during aging in the heart tissues, while *Il-1β* showed a significant reduction in *arg-ii-/-* animals compared to age-matched wild-type (*wt*) mice (Fig. 2). *Mcp1* is however a chemoattractant for macrophages and F4-80 serves as a pan marker for macrophages. Moreover, our previous studies demonstrate a relationship between Arg-II and IL-1β in vascular disease and obesity and age-associated renal and pulmonary fibrosis. Finally, IL-1β has been shown to play a causal role in patients with coronary atherosclerotic heart disease as shown by CANTOS trials. Therefore, we have focused on IL-1β in this study. We have now explained and strengthened this aspect in the manuscript on page 7, the last two lines and page 8, the 1st paragraph as following:

“Taking into account that our previous studies demonstrated a relationship of Arg-II and IL-1β in vascular disease and obesity (Ming et al., 2012) and in age-associated organ fibrosis such as renal and pulmonary fibrosis (Huang et al., 2021; Zhu et al., 2023), and IL-1β has been shown to play a causal role in patients with coronary atherosclerotic heart disease as shown by CANTOS trials (Ridker et al., 2017), we therefore focused on the role of IL-1β in crosstalk between macrophages and cardiac cells such as cardiomyocytes, fibroblasts and endothelial cells”.

(5) Although macrophages are shown to be involved in cardiac ageing in the arg-ii mouse model, the authors have not estimated macrophage infiltration and expression of inflammatory or senescence markers in the hearts of these mice.

Thank you very much for raising this important point! Taking the comments of the reviewer into account, we have performed new experiments, i.e., multiple immunofluorescent staining to analyze the infiltrated (CCR2^+^/F4-80^+^) and resident (LYVE1^+^/F4-80^+^) macrophage populations and to investigate to which extent that Arg-II affects the infiltrated and resident macrophage populations in the aging heart and whether this is regulated by *arg-ii*^*-/-*^. The results show an age-associated increase in the numbers of F4/80^+^ cells in the wt mouse heart, which is reduced in the age-matched *arg-ii*^*-/-*^ animals (Fig. 2G). This result is in accordance with the result of *f4/80* gene expression shown in Fig. 2A, demonstrating that *arg-ii* gene ablation reduces macrophage accumulation in the aging heart. Interestingly, resident macrophages as characterized by LYVE1^+^/F4-80^+^ cells (Fig. 2E and 2H) are predominant in the aging heart as compared to the infiltrated CCR2^+^/F4-80^+^ cells (Fig. 2F and 2I). The increase in both LYVE1^+^/F4-80^+^ and CCR2^+^/F4-80^+^ macrophages in aging heart is reduced in *arg-ii*^*-/-*^ mice (Fig. 2E, 2F, 2H, and 2I). These new results are described on page 6, the 1st paragraph, presented in Fig. 2E to 2I, and discussed on page 13, the 2nd, paragraph. The legend to Fig. 2 is revised. The method for this additional experiment is included on page 22, the 1st paragraph.

Moreover, the aged-associated accumulation of the senescence cells as demonstrated by p16^ink4^ positive cells is significantly reduced in *arg-ii-/-* animals. This new result is incorporated in the Fig. 1 as Fig. 1G and 1H and described / discussed on page 5, the 2nd paragraph and page 14, the 2nd last sentences of the 1st paragraph. The method of p16^ink4^ staining is included in the method section on page 22, the 1st paragraph, line 7. The legend to Fig. 1 is revised accordingly.

(6) Previously, Arg-II has been reported to serve a crucial role in ageing associated with reduced contractile function in rat hearts by regulating Nitric Oxide Synthase (PMID: 22160208). Elevated NO and superoxide have been shown to play crucial roles in the etiology of cardiovascular diseases (PMID: 24180388). Therefore, it is important to assess whether Nitric Oxide (NO) is involved in the aging-related phenotype in this mouse model.

Following the reviewer's suggestion, we conducted new experiments to investigate the role of nitric oxide (NO) in the context of the effect of Arg-II-induced IL-1b production in macrophages. We have addressed this question in the response to the comment point 2.

(7) Based on the results demonstrated in the study, ablation of Arg-II can be expected to cause a reduction in inflammation-associated phenotypes throughout the body at the multi-organ level. The observed improved cardiac phenotype could be an outcome of whole-body Arg-II ablation. It would be fruitful to develop a cardiac-specific Arg-II knockout mouse model to establish the role of Arg-II in the heart, independent of other organ systems.

We agree with the comment of the reviewer on this point. Unfortunately, as explained above (see point 1), it is currently not possible for us to perform the requested experiments, due to lack of cardiac specific *arg-ii*-knockout mouse model. Moreover, such an approach is complicated by the absence of Arg-II in cardiomyocytes and the expression of Arg-II in multiple cells including endothelial cells, fibroblasts and macrophage of different origin (resident and monocyte-derived infiltrating cells). It’s thus difficult to generate a cardiac-specific gene knockout mouse. One shall investigate roles of cell-specific Arg-II in cardiac aging by generating cell-specific *arg-ii-/-* mice. We appreciate very this important aspect and have discussed issue on page 19, the lines 2 to 6.

(8) Contrary to the findings in this paper, Arg-II has previously been reported to be essential for IL-10-mediated downregulation of pro-inflammatory cytokines, including IL-1β (PMID: 33674584).

Thank you very much for mentioning this study! We have now discussed thoroughly the controversies as the following on page 15, the last paragraph and page 16, the 1st paragraph;

“It is of note that a study reported that Arg-II is required for IL-10 mediated-inhibition of IL-1b in mouse BMDM upon LPS stimulation (Dowling et al., 2021), which suggests an anti-inflammatory function of Arg-II. The results of our present study, however, demonstrate that LPS enhances Arg-II and IL-1b levels in macrophages and knockout or silencing Arg-II reduces IL-1b production and release, demonstrating a pro-inflammatory effect of Arg-II. Our findings are supported by the study from another group, which shows decreased pro-inflammatory cytokine production including IL-6 and IL-1b in *arg-ii-/-* BMDM most likely through suppression of NFkB pathway, since *arg-ii-/-* BMDM reveals decreased activation of NFkB and IL-1b levels upon LPS stimulation (Uchida et al., 2023). Most importantly, our previous study also showed that re-introducing *arg-ii* gene back to the *arg-ii-/-* macrophages markedly enhances LPS-stimulated pro-inflammatory cytokine production (Ming et al., 2012), providing further evidence for a pro-inflammatory role of *arg-ii* under LPS stimulation. In support of this conclusion, chronic inflammatory diseases such as atherosclerosis and type 2 diabetes (Ming et al., 2012), inflammaging in lung (Zhu et al., 2023), kidney (Huang et al., 2021) and pancreas (Xiong, Yepuri, Necetin, et al., 2017) of aged animals or acute organ injury such as acute ischemic/reperfusion or cisplatin-induced renal injury are reduced in the *arg-ii-/-* mice (Uchida et al., 2023). The discrepant findings between these studies and that with IL-10 may implicate dichotomous functions of Arg-II in macrophages, depending on the experimental context or conditions. Nevertheless, our results strongly implicate a pro-inflammatory role of Arg-II in macrophages in the inflammaging in aging heart”.

(9) The authors have only performed immunofluorescence-based experiments to show fibrotic and apoptotic phenotypes throughout this study. To verify these findings, we suggest that they additionally perform RT-PCR or western blotting analysis for fibrotic markers and apoptotic markers.

The fibrotic aspect was analyzed not only by microscopy but also by using a quantitative biochemical assay such as hydroxyproline content assessment. Hydroxyproline is a major component of collagen and largely restricted to collagen. Therefore, the measurement of hydroxyproline levels can be used as an indicator of collagen content as previous investigated in the lung (Zhu et al., 2023). We have also measured collagen genes expression by RT-qPCR as suggested by the reviewer and found an age-related decline of collagen mRNA expression levels in both *wt* and *arg-ii-/-* mice, suggesting that the age-associated cardiac fibrosis and prevention in *arg-ii-/-* mice is due to alterations of translational and/or post-translational regulations, including collagen synthesis and/or degradation. The results are in accordance with that reported by other studies published in the literature. We have pointed out this aspect on page 5, the 2nd paragraph:

“The increased cardiac fibrosis in aging is however, associated with decreased mRNA levels of *collagen-Ia* (*col-Ia*) and *collagen-IIIa* (*col-IIIa*), the major isoforms of pre-collagen in the heart (Suppl. Fig. 2A and 2B), which is a well-known phenomenon in cardiac fibrotic remodelling (Besse et al., 1994; Horn et al., 2016). The results demonstrate that age-associated cardiac fibrosis and prevention in *arg-ii-/-* mice is due to alterations of translational and/or post-translational regulations including collagen synthesis and/or degradation”.

The results are presented in Suppl. Fig. 2, legend to Suppl. Fig. 2 is included in the file “Suppl. figure legend_R”. Suppl. table 2 for primers is revised accordingly.

We did not use additional markers to perform apoptotic assays with whole heart, since Fig. 3 shows good evidence that the aging is associated with increased apoptotic cells in the heart and significantly reduced in the *arg-ii-/-* mice. The reduction of TUNEL positive (apoptotic) cells in aged *arg-ii-/-* mice is mainly due to decrease in apoptotic cardiomyocytes. With the histological analysis, the apoptotic cell types can be well analysed. Moreover, biochemical assay for apoptosis such as caspase-3 cleavage with whole heart tissues can not distinguish apoptotic cell types and may not be sensitive enough for aging heart, due to relatively low numbers of apoptotic cells in aging heart as compared to myocardial infarct model.

(10) Figure 4: arg-ii has previously been reported to be expressed in rat cardiomyocytes (PMID: 16537391). We strongly suggest the authors verify the expression of Arg-II via immunostaining in isolated cardiomyocytes (using published protocols), and by using multiple different cardiomyocyte-specific markers for colocalization studies to prove the lack of arg-ii expression beyond a reasonable doubt.

As pointed out by this reviewer, Arg-II was previously reported to be expressed in isolated cardiomyocytes from in rats (PMID: 16537391). Unfortunately, negative controls. i.e., *arg-ii-/-* samples were not included in the study to avoid any possible background signals. We made great effort to investigate whether Arg-II is present in the cardiomyocytes from different species including mice, rats and humans and have included old *arg-ii-/-* mouse samples as a negative control. This allows to validate the antibody specificity and background noises beyond any reasonable doubt. The new experiments in Suppl. Fig. 4 confirms the specificity of the antibody against Arg-II in old mouse kidney which is known to express Arg-II in the S3 proximal tubular cells (Huang J, et al. 2021). To exclude the possible species-specific different expression of Arg-II in the cardiomyocytes, aged mouse and rat heart tissues were used for cellular localization of Arg-II by confocal immunofluorescence staining. As shown in Suppl. Fig. 4B and 4C, both species show Arg-II expression only in non-cardiomyocytes (cells between striated cardiomyocytes) (red arrows) but not in striated cardiomyocytes. Even in the rat myocardial infarction tissues, Arg-II was not found in cardiomyocytes but in endocardium cells (Suppl. Fig. 4B). In isolated cardiomyocytes exposed to hypoxia, a well know strong stimulus for Arg-II protein levels, no Arg-II signals could be detected, while in fibroblasts from the same animals, an elevated Arg-II levels under hypoxia is demonstrated (Fig. 5B). Furthermore, RT-qPCR could not detect *arg-ii* mRNA in cardiomyocytes but in non-cardiomyocytes (Fig. 5C). All together, these results demonstrate that Arg-II are not expressed or at negligible levels in cardiomyocytes but expressed in non-cardiomyocytes. This new experiments with rat heart are included in the method section on page 20, the 1st paragraph. The results are described on page 7, the 1st paragraph, and discussed on page 12, the 2nd paragraph. Legend to Suppl. Fig. 4 is included in the file “Suppl. figure legend_R”.

(11) Figure 6G: It may be worthwhile to supplement *arg-ii-/-* old cells with IL-1beta to see if there is an increase in TUNEL-positive cells.

IL-1b is a well known pro-inflammatory cytokine that causes apoptosis in various cell types including cardiomyocytes (Shen Y., et al., Tex Heart Inst J. 2015;42:109–116. doi: 10.14503/THIJ-14-4254; Liu Z. et. al., Cardiovasc Diabetol 2015;14,125. doi: 10.1186/s12933-015-0288-y; Li. Z., et al., Sci Adv 2020;6:eaay0589. doi: 10.1126/sciadv.aay0589). We appreciate very much the interesting idea of this reviewer to investigate the apoptotic responses of cardiomyocytes from *arg-ii-/-* mice to IL-1b. We agree that it is possible that cardiomyocytes from *wt* from *arg-ii-/-* mice react differently to IL-1b, although the cardiomyocytes do not express Arg-II as demonstrated in our present study. If this is true, it must be due to non-cell autonomous effects of different aging microenvironment in the heart or epigenetic modulations of the myocytes. We found that this is a very interesting aspect and requires further extensive investigation. Since our current study focused on the effect of *wt* and *arg-ii-/-* macrophages on cardiomyocytes and non-cardiomyocytes, we prefer not to include this suggested aspect in our manuscript and would like to explore it in the following study.

(12) Figures 4-9: It would be interesting to see if the effect of ArgII in cardiac ageing is gender-specific. It is recommended to include experimental data with male mice in addition to the results demonstrated in female mice.

As pointed out in the manuscript, we have focused on female mice, because an age-associated increase in *arg-ii* expression is more pronounced in females than in males (Fig. 1A). As suggested by this reviewer, we performed additional experiments investigating effects of *arg-ii* deficiency in male mice during aging, focusing on pathophysiological outcomes of ischemia/reperfusion injury in ex vivo experiments. The ex vivo functional analytic experiments with Langendorff system were performed in aged male mice (see Suppl. Fig. 9). Following ischemia/reperfusion injury, *wt* male mice display reduced left ventricular developed pressure (LVDP), as well as the inotropic and lusitropic states (expressed as dP/dt max and dP/dt min, respectively). As previously reported (Murphy et al., 2007), we also found that old male mice are more prone to I/R injury than age-matched female animals. Specifically, 15 minutes of ischemia are enough to significantly affect the left ventricle contractile function in the male mice (Suppl. Fig. 9). As opposite, age-matched old female mice are relatively resistant to I/R injury, and at least 20 min of ischemia are necessary to induce a significant impairment of the contractile function (Fig. 10). Similar to females, the post I/R recovery of cardiac function is also significantly improved in the male *arg-ii-/-* mice as compared to age-matched *wt* animals. In addition to functional recovery, triphenyl tetrazolium chloride (TTC) staining (myocardial infarction) upon I/R-injury in males is significantly reduced in the age-matched male *arg-ii-/-* animals (Suppl. Fig. 9C and 9D). All together, these results reveal a role for Arg-II in heart function impairment during aging in both genders with a higher vulnerability to stress in the males. These new results are presented in Suppl. Fig. 9, described on page 10, the last paragraph and page 11. The results are discussed on page 18, the 2nd paragraph as following:

“The fact that aged females have higher Arg-II but are more resistant to I/R injury seems contradictory to the detrimental effect of Arg-II in I/R injury. It is presumable that cardiac vulnerability to injuries stressors depends on multiple factors/mechanisms in aging. Other factors/mechanisms associated with sex may prevail and determine the higher sensitivity of male heart to I/R injury, which requires further investigation. Nevertheless, the results of our study show that Arg-II plays a role in cardiac I/R injury also in males”.

The information on the experimental methods in the male animals is included on page 20, the last paragraph and page 21, the 1st paragraph. Legend to Suppl. Fig. 9 is included in the file “Suppl. figure legend_R”.

(13) Figure 6G: cardiomyocytes from wild-type mice, when treated with macrophages, show 0% TUNEL-positive cells. Since it is unlikely to obtain no TUNEL staining in a cell population, there may be an experimental or analytical error.

Now it is Fig. 7F and 7G. This is due to our specific experimental procedure. After tissue digestion, cardiomyocytes were plated on laminin-coated dishes. Laminin promotes the adhesion of survived cells. Following plating, we conducted a deep washing process to remove damaged and partially adherent cells. This step ensures that only well-shaped, viable, and strongly adherent cells remain as bioassay cells. These “healthy” cells are then selected for the experiments. the apoptotic cells are removed by washing out, reflecting the high viability of the bioassay cells. We have added this detailed information in the method section on page 24, the 2nd paragraph.

(14) Figure 7J: Please assess whether arg-ii depletion also affects the mtROS phenotype.

According to the suggestion of this reviewer, we performed new experiments which show that human cardiac fibroblasts (HCFs) exposed to hypoxia (1% O_2_, 48 hours), a known physiological trigger of Arg-II up-regulation, exhibit increased mtROS generation, which involves Arg-II (new Fig. 8M to 8P). We found that Arg-II protein level as well as mtROS (assessed by mitoSOX staining) were both enhanced, accompanied by increased levels of HIF1α (Fig 8M). Moreover, mito-TEMPO pre-incubation reduces mtROS, confirming the mitochondrial origin of the ROS. Silencing of *arg-ii* with rAd-mediated shRNA, significantly reduces mtROS levels demonstrating a role of Arg-II in the production of mitochondrial ROS in cardiac fibroblasts (Fig 8M to 8P). We have included these results on page 9, the last paragraph and discussed the results on page 17, the 1st paragraph. The related method is described on page 26, the 2nd paragraph. Legend to Fig. 8 is updated on page 32.

(15) Figure 8A-E: The authors have treated human-origin endothelial cells with mice-origin macrophage-conditioned media. It would be more suitable to treat the endothelial cells with human-origin macrophage-conditioned media.

We acknowledge the concern regarding the use of mouse-origin macrophage-conditioned media on human-origin endothelial cells. It is to note, the biological cross-reactivity of cytokines from one species on cells from a different species has been reported in the literature. It was observed that there is quite a strict threshold of 60% amino acid identity, above which cytokines tend to cross-react and statistically, cytokines would tend to cross-react more often as their % amino acid identity increases (Scheerlinck JPY. Functional and structural comparison of cytokines in different species. Vet Immunol Immunopathol. 1999; 72:39-44. https://doi.org/10.1016/S0165-2427(99)00115-4). Taking IL-1b as an example, the 17.5 kDa mature mouse and human IL-1b share 92% aa sequence identity, suggesting a high cross-reactivity. Indeed, human IL-1b has shown biological cross-reactivity in mouse cells (Ledesma E., et al. Interleukin-1 beta (IL-1β) induces tumor necrosis factor alpha (TNF-α) expression on mouse myeloid multipotent cell line 32D cl3 and inhibits their proliferation. Cytokine. 2004; 26:66-72. https://doi.org/10.1016/j.cyto.2003.12.009). Moreover, our results also support the reported cross-reactivity between human and mouse IL-1b. The CM from mouse macrophage indeed showed biological function in human endothelial cells. The observed effects of the conditioned media from aged wild-type macrophages on endothelial cells were specifically mediated through IL-1β. This conclusion is supported by our data showing that the upregulation induced by the conditioned media was significantly reduced by the addition of an IL-1β receptor blocker.

(16) The co-culture system would be more interesting to test the non-cell autonomous role of Arg II.

We appreciate the suggestion by this reviewer regarding the co-culture system to test the non-cell autonomous role of Arg-II. We believe that our current model, which involves treating cells with conditioned media, is a well-established and effective method for demonstrating the non-cell autonomous role of Arg-II. This approach allows us to observe the effects of Arg-II on surrounding cells through the factors present in the conditioned media. The co-culture system could be considered, if the released factor in the conditioned medium is not stable. This is however not the case. So we are confident that our experimental model with conditioned medium is good enough to demonstrate a paracrine effect of cell-cell interaction.

**Reviewer #2 (Recommendations For The Authors):**
Some minor comments may be considered to improve the realm of the knowledge related to this study.

We appreciate this comment and have added and revised our discussion on this aspect accordingly at the end of the discussion section on page 19, the last 6 lines.

(1) The current study showed strong evidence demonstrating the key role of cardiac macrophages in pathologies of cardiac aging, particularly, the macrophages (MФ) from the circulating blood (hematogenous). It is known that the heart is among the minority of organs in which substantial numbers of yolk-sac MФ persist in adulthood and play a crucial role in maintaining cardiac function. Thus, the adult mammalian heart contains two separate and discrete cardiac MФ subgroups, i.e., the resident MФs originated from yolk sac-derived progenitors and the hematogenous MФs recruited from circulating blood monocytes. These two subtypes of MФs may play distinctive roles in the aging heart and the response to cardiac injury. The author could extend the discussion on the possibility of the resident MФs in aging hearts, which could be further investigated in the future.

We appreciate the suggestion and agree that it provides valuable insight into the study. Taking the comments of the reviewer 1 into account, we have performed new experiments, i.e., co- immunostaining to analyze the infiltrated (CCR2^+^/F4-80^+^) and resident (LYVE1^+^/F4-80^+^) macrophage populations and to investigate to which extent that Arg-II affects infiltrated and resident macrophage populations in the aging heart. We found that in line with the gene expression of *f4/80*, immunofluorescence staining reveals an age-associated increase in the numbers of F4/80^+^ cells in the *wt* mouse heart, which is reduced in the age-matched *arg-ii-/-* animals (Fig. 2E, F, G), demonstrating that *arg-ii* gene ablation reduces macrophage accumulation in the aging heart. Interestingly, resident macrophages as characterized by LYVE1^+^/F4-80^+^ cells (Fig. 2E and 2H) are predominant in the aging heart as compared to the infiltrated CCR2^+^/F4-80^+^ cells (Fig. 2F and 2I). The increase in both LYVE1^+^/F4-80^+^ and CCR2^+^/F4-80^+^ macrophages in aging heart is reduced in *arg-ii-/-* mice (Fig. 2E, 2F, 2H, and 2I). These new results are described on page 6, the 1st paragraph, presented in Fig. 2E to 2I, and discussed on page 13, the 2nd, paragraph. The legend to Fig. 2 is revised. The method for this additional experiment is included on page 22, the 1st paragraph.

(2) It would be beneficial to the readers if the author could provide some explanation about why ArgII could not be detected in VSMCs in the mouse heart and the species difference between humans and mice. In addition, the author may provide an assumption on the possibility that there may also be a cross-talk between macrophages and VSMCs in the aging heart. A little bit more explanation in the Discussion will be helpful.

We acknowledge and appreciate the suggestion and have discussed these points on page 19 as the following:

“In this context, another interesting aspect is the cross-talk between macrophages and vascular SMC in the aging heart. In our present study, we could not detect Arg-II in vascular SMC of mouse heart but in that of human heart. This could be due to the difference in species-specific Arg-II expression in the heart or related to the disease conditions in human heart which is harvested from patients with cardiovascular diseases. Indeed, in the *apoe-/-* mouse atherosclerosis model, aortic SMCs do express Arg-II (Xiong et al., 2013). It is interesting to note that rodents hardly develop atherosclerosis as compared to humans. Whether this could be partly contributed by the different expression of Arg-II in vascular SMC between rodents and humans requires further investigation. In our present study, the aspect of the cross-talk between macrophages and vascular SMC is not studied. Since the crosstalk between macrophages and vascular SMC has been implicated in the context of atherogenesis as reviewed (Gong et al., 2025), further work shall investigate whether Arg-II expressing macrophages could interact with vascular SMC in the coronary arteries in the heart and contribute to the development of coronary artery disease and/or vascular remodelling and the underlying mechanisms“.

(3) Please clarify the arrows in Figure 9C that indicate the infarct area in each splicing section from one heart.

The arrows in Figure 9C (now Fig. 10C) are indeed utilized to indicate the sections displaying the infarcted area within each splicing section from one heart. We have explained the arrow in the figure legend (now Fig. 10 and also new Suppl. Fig. 9).